# Mixed model-based deconvolution of cell-state abundances (MeDuSA) along a one-dimensional trajectory

Liyang Song[1,2,3], Xiwei Sun[2,3], Ting Qi[2,3] & Jian Yang ◉ [2,3] ✉

Deconvoluting cell-state abundances from bulk RNA-sequencing data can add considerable value to existing data, but achieving fine-resolution and high-accuracy deconvolution remains a challenge. Here we introduce MeDuSA, a mixed model-based method that leverages single-cell RNA-sequencing data as a reference to estimate cell-state abundances along a one-dimensional trajectory in bulk RNA-sequencing data. The advantage of MeDuSA lies primarily in estimating cell abundance in each state while fitting the remaining cells of the same type individually as random effects. Extensive simulations and real-data benchmark analyses demonstrate that MeDuSA greatly improves the estimation accuracy over existing methods for one-dimensional trajectories. Applying MeDuSA to cohort-level RNA-sequencing datasets reveals associations of cell-state abundances with disease or treatment conditions and cell-state-dependent genetic control of transcription. Our study provides a high-accuracy and fine-resolution method for cell-state deconvolution along a one-dimensional trajectory and demonstrates its utility in characterizing the dynamics of cell states in various biological processes.

Cellular deconvolution is a computational technique aimed to estimate cellular compositions from tissue-level 'bulk' omics data[1,2]. With the increasing availability of bulk RNA-sequencing (RNA-seq) data, cellular deconvolution has become a pivotal approach for estimating cell-type compositions in a tissue of interest. This methodological advance has greatly facilitated research to understand the roles of different cell types in dynamic disease processes (for example, quantifying immune cell infiltrations in solid tumors)[3–5], probe genetic regulatory mechanisms at the cellular level (for example, cell-type-specific expression quantitative trait locus analysis)[6–9] and adjust biases caused by cell-type compositions in association analyses (for example, using cell-type compositions for covariate adjustment)[7,10,11].

Over the past decade, many cellular deconvolution methods have been developed and benchmarked[1,2], including BayesPrism[12], CIBERSORT[13] and MuSiC[14] among others. Most of them share a typical workflow, that is, generating cell-type-specific gene expression profiles (GEPs) from a reference, such as bulk RNA-seq data from individual cell subsets (for example, CIBERSORT[13]) or single-cell RNA-seq (scRNA-seq) data (for example, MuSiC[14]), and utilizing the reference GEPs to compute cell-type compositions in bulk RNA-seq data. Nevertheless, cells of the same type are not homogeneous but distributed across multiple states in a biological process that arises in a context-dependent manner, for example, activation[15], differentiation[16] or degeneration[17]. This distribution can vary among different environments, disease conditions and genetically distinct individuals. In this regard, further opportunities and challenges of cellular deconvolution lie in estimating the abundances of cells at different states (referred to as cell-state abundance hereafter) in bulk RNA-seq data.

Single-cell RNA-seq offers a snapshot of the transcriptome of thousands of diverse cells, providing an avenue for studying cell states in various biological processes[18,19]. In scRNA-seq data, cells at different states can be computationally ordered to infer cell-state trajectories

[1]College of Life Sciences, Zhejiang University, Hangzhou, Zhejiang, China. [2]School of Life Sciences, Westlake University, Hangzhou, Zhejiang, China. [3]Westlake Laboratory of Life Sciences and Biomedicine, Hangzhou, Zhejiang, China. ✉e-mail: jian.yang@westlake.edu.cn

(for example, pseudotime)[19]. Cell population mapping (CPM)[20] is a cellular deconvolution method specifically designed to exploit 'cell-state space' inferred from reference scRNA-seq data to estimate cell-state abundances in bulk RNA-seq data. CPM partitions the cell-state space into several grids, constructs a GEP by randomly sampling a cell from each grid and combines the estimated abundances across thousands of repeats to obtain a single abundance for each cell. While CPM has considerably improved the deconvolution resolution, the accuracy of the estimated cell-state abundance can still be improved, largely because it focuses on only a small number of cells in each sampling repeat without accounting for the remaining cells.

In this Article, we introduce MeDuSA (mixed model-based deconvolution of cell-state abundances), a high-accuracy and fine-resolution cellular deconvolution method that leverages scRNA-seq data as a reference to estimate cell-state abundances along a one-dimensional trajectory in bulk RNA-seq data. MeDuSA features the use of a linear mixed model (LMM) to fit a cell state in question (either a single cell or the mean of multiple cells, referred to as the focal state hereafter) as a fixed effect and the remaining cells of the same cell type individually as random effects accounting for correlations between cells. This model improves the deconvolution accuracy because the random-effect component allows each cell to have a specific weight on bulk gene expression, resulting in a better capturing of variance in bulk gene expression, and ameliorates the collinearity problem between the cell(s) at the focal state (fitted as a fixed effect) and those at adjacent states (fitted as random effects). We show by extensive simulations and real-data benchmark analyses that MeDuSA is substantially more accurate than existing methods when assessed with one-dimensional trajectories. We also demonstrate the utility of MeDuSA by applying it to cohort-level bulk RNA-seq data to reveal associations of cell-state abundances with disease or treatment conditions and cell-state-dependent genetic control of transcription.

## Results

### Simulation study

The MeDuSA method is described in Methods, with a schematic summary shown in Fig. 1 and the technical details presented in sections 1 and 2 of the Supplementary Note. Briefly, MeDuSA utilizes scRNA-seq data as a reference to estimate the abundance of cells at various states along a one-dimensional trajectory in bulk RNA-seq data. This is done using an LMM in which the focal state is fitted as a fixed effect and cells at the other states are fitted individually as random effects, with the other cell types fitted as fixed covariates. We performed a series of simulations to assess the performance of MeDuSA and evaluate the robustness of MeDuSA to the choice of parameters ('Simulation strategy' in Methods). Our simulations were based on 17 scRNA-seq datasets generated from different species and sequencing platforms with varying number of cells captured (Supplementary Table 1). The cell types and cell-state trajectories of these datasets were annotated and validated previously or in this study (Methods). We split each scRNA-seq dataset into two portions, randomly assigning one portion as simulation source data and the other portion as deconvolution analysis reference data. Synthetic bulk RNA-seq data were generated as mixtures of scRNA-seq profiles, according to four pre-designed cell-state distribution patterns (Fig. 2a). We compared MeDuSA with CPM[20], along with cell-type deconvolution and gene enrichment-based methods, including BayesPrism[12], MuSiC[14], CIBERSORT[13], Scaden[21], TAPE[22] and ssGSEA[23], that can be repurposed for cell-state deconvolution by dividing the cell-state trajectory into cell bins (section 3 of the Supplementary Note). The deconvolution accuracy was measured by the concordance correlation coefficient (CCC), Pearson's correlation ($R$) and the root mean square deviation (RMSD) between the estimated cell-state abundance and the ground truth. We used CCC as the primary measure of deconvolution accuracy, as it is less sensitive to overweighted outliers than $R$ and more interpretable than RMSD.

The results showed that MeDuSA outperformed the compared methods by a considerable margin for one-dimensional trajectories, especially when the cell-state abundance distribution was non-monotonic (Fig. 2b and Supplementary Figs. 1 and 2). For instance, when the distribution was bimodal, the deconvolution accuracy of MeDuSA (CCC) was 0.85, 3.4-fold higher than the best-performing methods among CPM (−0.05), BayesPrism (0.15), MuSiC (0.25), CIBERSORT (0.13), Scaden (0.03), TAPE(0.007) and ssGSEA (0.23).

We performed a series of sensitivity analyses to investigate the factors that influence the performance of MeDuSA (or cellular deconvolution in general). First, smoothing slightly improved the deconvolution accuracy of MeDuSA (from 0.76 to 0.86), despite that MeDuSA without smoothing (MeDuSA-NS) performed considerably better than the other methods including CPM with smoothing (Supplementary Fig. 2). As smoothing can mask the effects of the other factors, we performed the sensitivity analyses below without the smoothing step. Second, the random-effect component was nominally significant ($P < 0.05$) in all simulations (maximum $P = 1.35 \times 10^{-10}$), and the significance level decreased with fewer cells fitted (Supplementary Fig. 3). The accuracy of MeDuSA-NS also decreased with fewer cells fitted in the random-effect component (Extended Data Fig. 1), suggesting the benefit of fitting all cells at non-focal states as random effects to reduce residual variance. Third, instead of fitting the non-focal cells each as a random effect, we grouped them into bins along the cell-state trajectory and fitted the mean of each bin as a random effect; the accuracy decreased dramatically from 0.76 to 0.33 (Supplementary Fig. 4), demonstrating the benefit of allowing each cell to have a specific weight on bulk gene expression. Fourth, the accuracy decreased to 0.17 when we fitted the bins each as a fixed effect (Supplementary Fig. 5), showing the benefit of fitting the non-focal states as random effects (to ameliorate the collinearity problem between the focal and non-focal states), as further evidenced by the increased difference between the fixed- and random-effect models with the level of collinearity (Supplementary Fig. 6). Fifth, ignoring the correlations between cells in the random-effect component resulted in decreased deconvolution accuracy (from 0.75 to 0.63), especially when the underlying cell-state abundance distribution is complex (Supplementary Fig. 7). Sixth, we showed that the accuracy of MeDuSA was generally robust when the number of cell states varied from 50 to 1,000 (a larger number of cell states representing a higher deconvolution resolution) (Supplementary Figs. 8 and 9). Finally, we demonstrated the confounding effects of other cell types, which could largely be corrected by fitting the mean expression of each of them as a fixed-effect covariate (Supplementary Fig. 10).

While the use of the LMM improves the deconvolution accuracy, it introduces a much higher level of computational complexity than the models used in CPM and other cell-type deconvolution methods. We improved the computational efficiency of MeDuSA through coding the core algorithm with C++ and applying an appropriate approximation algorithm ('Computational speed-up' in Methods). On a unified computing platform with one central processing unit, the runtime of MeDuSA to deconvolute a bulk RNA-seq dataset using a Smart-seq2 or 10X Genomics scRNA-seq dataset as the reference (10,000 cells in both datasets) was 17.1 min or 5.6 min, respectively, 5.3-fold or 3.3-fold faster than CPM (Extended Data Fig. 2).

### Benchmark analysis with real bulk RNA-seq data

We then benchmarked the performance of the deconvolution methods with real bulk RNA-seq data. Four sample-matched scRNA-seq and bulk RNA-seq datasets from human esophagi ($n = 15$), human bone marrows ($n = 8$), induced pluripotent stem cells (iPSCs; $n = 6$) and human embryonic stem cells (hPSCs; $n = 6$) were used in this analysis (Supplementary Fig. 11). In real data, the true cell-state abundances are unknown and need to be estimated. In each dataset, we inferred the cell-state trajectory and estimated the corresponding

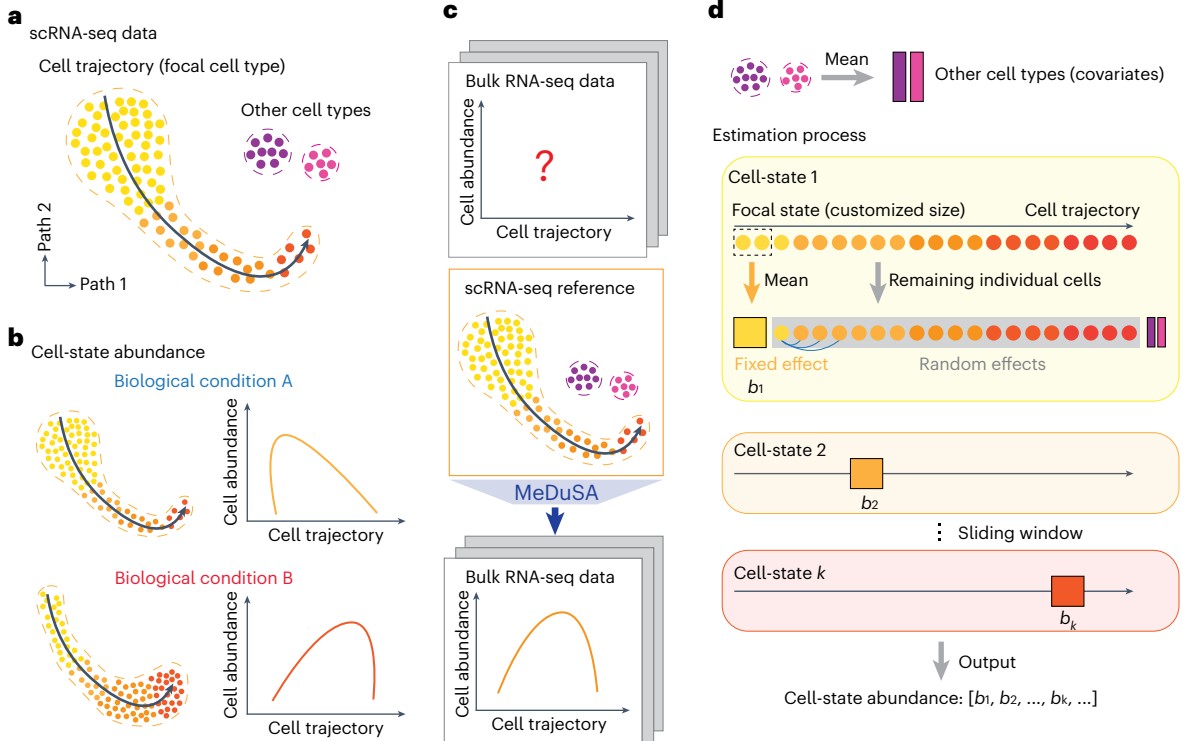

**Fig. 1 | Schematics of the concept of cell-state trajectory deconvolution and the MeDuSA model. a**, Cells colored in orange are ordered by the cell-state trajectory. **b**, The distribution of cells along the cell-state trajectory, also known as cell-state abundance distribution, varies under different biological conditions. This distribution can be profiled in scRNA-seq data but is not directly achievable in bulk RNA-seq data. **c**, MeDuSA is a fine-resolution cellular deconvolution method that leverages scRNA-seq data as a reference to estimate cell-state abundance in bulk RNA-seq data. **d**, An overview of the cell-state abundance estimation process. Briefly, MeDuSA fits the focal cell-state as the fixed effect, while simultaneously fitting the remaining cells along the trajectory individually as random effects. $b_k$ represents the abundance of cells at state $k$. Further details regarding the MeDuSA method can be found in Methods and sections 1 and 2 of the Supplementary Note.

cell-state abundances from scRNA-seq data. We then compared the estimated cell-state abundances with those obtained from bulk RNA-seq data using the deconvolution methods (Fig. 3a,b). We performed cross-validation where applicable, that is, the samples used for cell-state trajectory inference were excluded from the deconvolution analysis. The results again showed that MeDuSA substantially outperformed the compared methods (Fig. 3c and Supplementary Fig. 12). The mean deconvolution accuracy (CCC) of MeDuSA was 0.70, 2.2-fold higher than the best-performing method among CPM (0.31), BayesPrism (0.19), MuSiC (0.07), CIBERSORT (0.08), Scaden (0.06), TAPE (0.07) and ssGSEA (0.17) (Fig. 3c). The conclusion remained mostly consistent at higher deconvolution resolutions (Extended Data Fig. 3). It is noteworthy that the performances of the deconvolution methods in this real-data benchmark analysis were generally lower than those in the simulation study, probably because of the discrepancies between the bulk RNA-seq and scRNA-seq (Supplementary Fig. 13) and the uncertainty in estimating cell-state abundances from scRNA-seq data[19].

We further compared the cell-state abundances of epithelia estimated from the esophagus to those estimated from other tissues without the keratinization process, which can be regarded as negative controls. We applied MeDuSA to bulk RNA-seq data from esophagus mucosa ($n = 555$), blood ($n = 929$), heart ($n = 861$), liver ($n = 226$), spleen ($n = 241$), colon ($n = 779$) and small intestine ($n = 187$) in the Genotype-Tissue Expression (GTEx), using the fresh esophageal scRNA-seq data above as the reference. Compared with the abundance of epithelium estimated from esophagus, the abundance of epithelium estimated from the non-esophageal tissues was small (Fig. 3d).

## Case studies

We next applied MeDuSA in four case studies to demonstrate how a cell-state abundance deconvolution method with substantially improved accuracy can give rise to deeper insights into disease etiology and biological mechanisms.

### Application to esophageal carcinoma

We applied MeDuSA to conduct cell-state abundance deconvolution analyses in 109 human esophagus bulk RNA-seq data from The Cancer Genome Atlas (TCGA), of which 98 samples were collected from the esophageal squamous-cell carcinoma (ESCC) tumor tissue, and 11 samples were collected from the adjacent normal esophageal tissue, using the scRNA-seq data from the normal fresh esophageal tissue above as the reference. In this reference data, cell types were annotated according to the marker genes, and the keratinization trajectory of epithelial cells was inferred using Slingshot[24] (Fig. 4a). The keratinization trajectory profiles the cytodifferentiation process of epithelial cells, proceeding from the post-germinative state (that is, the basal layer of the epithelium) to the finally cuticularized state (that is, the upper layer of the epithelium) (Fig. 4b,c). ESCC arises from the basal layer of the esophagus epithelium, resulting in a thicker basal layer than that in normal esophagi[25,26]. Hence, the abundance of epithelial cells in the basal layer (that is, in the earlier stage of the keratinization trajectory) in tumor is expected to be higher than that in normal esophagi. Such an expected histological change can be detected by MeDuSA, as evidenced by the significant difference in the abundance distribution of epithelial cells over the keratinization trajectory between ESCC and normal esophagi (permutation *F*-test, $P = 0.012$; Fig. 4d and 'Testing

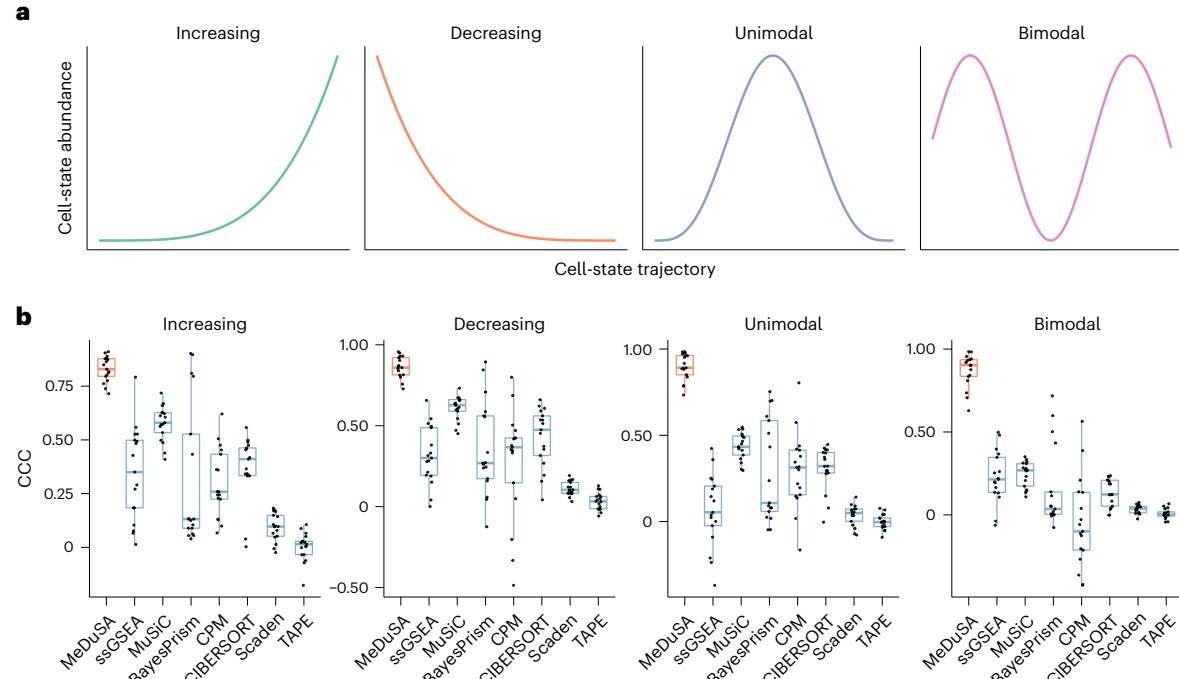

**Fig. 2 | Benchmarking the cellular deconvolution methods by simulations.**
**a**, Pre-designed distributions of cell abundance along the cell-state trajectory.
We generated synthetic bulk RNA-seq data as mixtures of scRNA-seq profiles
according to each of the four pre-designed cell-state abundance distributions
('Simulation strategy' in Methods). **b**, Boxplot of CCC (the higher the better)
for each deconvolution method. Each dot represents the mean deconvolution
accuracy over five replicates for a simulation source dataset. The box indicates
the interquartile range (IQR), the line within the box represents the median value
and the whiskers extend to data points within 1.5 times the IQR.

for differences in cell-state abundances among groups' in Methods).
Considering that the difference was only marginally significant, prob-
ably due to the small sample size of normal esophagi in TCGA ($n = 11$),
we combined TCGA data with the data to increase the sample size of
normal esophagi to 664. After adjusting for batch effects between
TCGA and GTEx[27] (Supplementary Fig. 14), we observed similar result
as above that relative to normal esophagi, abundance of epithelial
cells inferred from tumor tissues shifted toward the basal layer (per-
mutation $F$-test, $P < 1 \times 10^{-4}$, with the $P$ value capped by the number of
permutations; Fig. 4e). An accordant result was obtained in another
independent esophagus bulk RNA-seq dataset ($n = 46$, permutation
$F$-test, $P = 3.2 \times 10^{-4}$; Fig. 4f), using another independent scRNA-seq
dataset as the reference (Supplementary Fig. 15).

**Application to COVID-19**
We applied MeDuSA to RNA-seq data from patients with coronavirus
disease 2019 (COVID-19), with the aim to portray the dynamics of CD8+ T
cells during the severe acute respiratory syndrome coronavirus 2 (SARS-
CoV-2) infection. A COVID-19 peripheral blood mononuclear cell (PBMC)
scRNA-seq dataset from 6 healthy and 7 SARS-CoV-2-infected donors
was used as the reference for the deconvolution analyses (Extended
Data Fig. 4a). Altogether, we retrieved 6,762 CD8+ T cells, which were
then classified into three subtypes, including naive T cells (T$_n$), effector-
memory T cells (T$_{em}$) and exhaustion-like T cells (T$_{ex}$). The gamma delta
T cells were excluded as their development process is disjoint from the
other subtypes of CD8+ T cells. Diffusion map and RNA velocity analyses
suggested that the CD8+ T cells developed from the naive state (T$_n$) to the
exhaustion state (T$_{ex}$) (Extended Data Fig. 4b and Supplementary Fig. 16),
as validated by the expression pattern of the marker genes (Extended
Data Fig. 4c), consistent with previous studies[28,29].

Using the reference scRNA-seq above, we deconvoluted a PBMC
bulk RNA-seq dataset consisting of 17 healthy donors and 17 patients

with COVID-19. We observed a significant difference in the abun-
dance distribution of CD8+ T cells over the development trajectory
between healthy donors and patients with COVID-19 (Extended Data
Fig. 4d). Compared with healthy donors, CD8+ T cells from patients
with COVID-19 were enriched in the exhaustion state (permutation
$F$-test, $P = 1.4 \times 10^{-3}$), in line with previous studies[28]. These results were
replicated in another independent COVID-19 bulk RNA-seq dataset,
comprising 10 healthy donors and 44 patients with COVID-19 (permuta-
tion $F$-test, $P = 2.6 \times 10^{-4}$; Extended Data Fig. 4e).

To further investigate the variation of CD8+ T cells among patients
with COVID-19 under different clinical conditions, we applied MeDuSA
to another PBMC bulk RNA-seq dataset containing 100 patients with
relevant clinical indicators. After grouping patients into tertiles
according to their blood C-reactive protein (CRP) levels, we found
that CD8+ T cells from patients with higher CRP levels showed higher
enrichment in the exhaustion state (permutation $F$-test, $P = 0.038$;
Extended Data Fig. 4f), supporting the hypothesis that inflamma-
tion-associated stress may contribute to the dysregulation of CD8+
T cells in patients with COVID-19[29]. We further analyzed a bulk RNA-
seq COVID-19 dataset from patients under different World Health
Organization scored clinical phases (13 patients with COVID-19 and
14 healthy donors) (Extended Data Fig. 4h). The result showed a clear
trend that patients with COVID-19 at convalescence stages (that is,
clinical phases 6 and 7) had similar abundance distribution of CD8+ T
cells over the development trajectory to healthy donors (clinical phase
0); in contrast, patients at disease stages (that is, clinical phases 1–5)
tended to aggregate together, showing enrichment of CD8+ T cells in
the high-exhaustion state (Extended Data Fig. 4g). In summary, our
results revealed the dynamics of cell-state abundance of CD8+ T cells
over the development trajectory during the SARS-CoV-2 infection,
suggesting that CD8+ T cells in patients with COVID-19 were enriched
in the inflammation-associated exhaustion state.

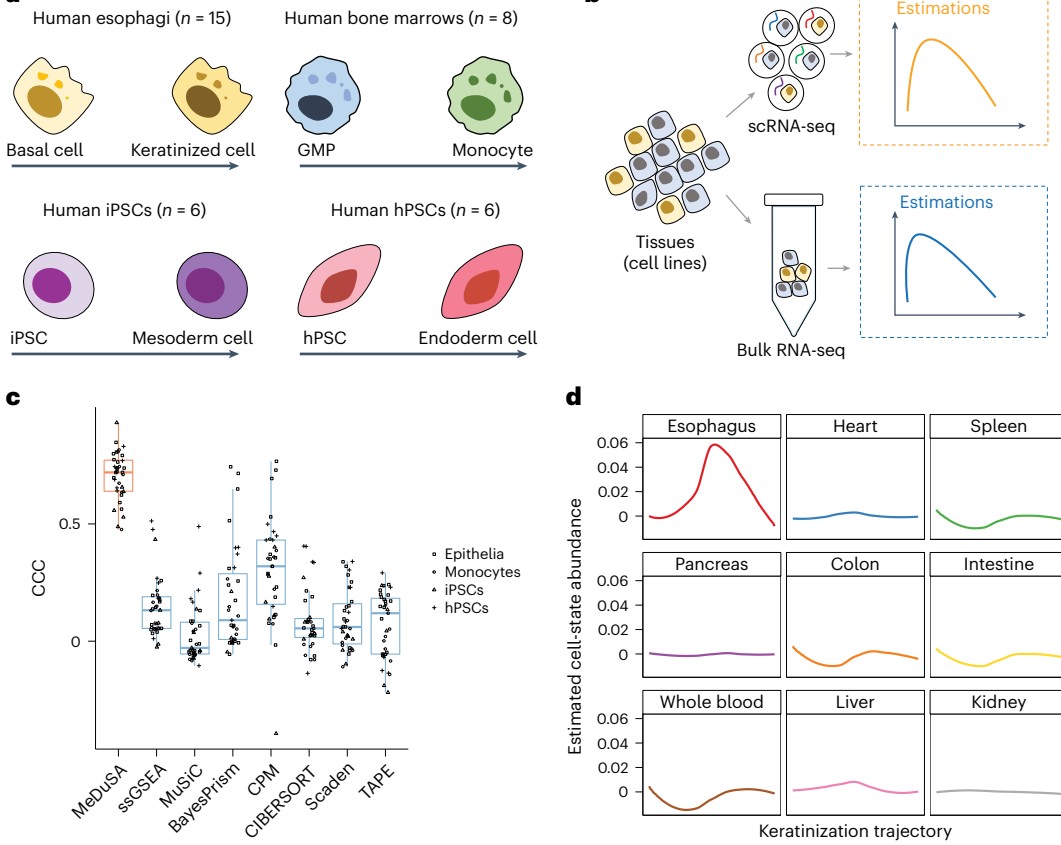

**Fig. 3 | Benchmarking the cellular deconvolution methods by the analysis of real bulk RNA-seq data. a,b,** Schematic overview of the included datasets (**a**) and experiment design (**b**) in real-data benchmark analysis. GMP, granulocyte-monocyte progenitor cells. **c,** Boxplot of CCC for each deconvolution method. We benchmarked the performance of the methods using sample-matched bulk RNA-seq and scRNA-seq data from four different tissues or cell lines (n = 35). Each dot represents the deconvolution accuracy (measured by CCC) for a sample, with

the shape indicating the corresponding tissue or cell line. The box indicates the IQR, the line within the box represents the median value and the whiskers extend to data points within 1.5 times the IQR. **d,** The estimated epithelia abundance along the keratinization trajectory in the GTEx tissues. The x axis represents the keratinization trajectory, and the curved line shows the mean cell-state abundance across individuals.

## Application to skin melanoma

A previous scRNA-seq study of skin melanoma shows that low-exhaustion CD8[+] T cells are depleted in T-cell receptor (TCR) expanded clusters but enriched in TCR non-expanded clusters[30]. In other words, TCR clonal expansions might be positively correlated with the exhaustion state of CD8[+] T cells. Using this melanoma scRNA-seq dataset as the reference (Fig. 5a), we applied MeDuSA to TCGA melanoma bulk RNA-seq data (n = 430). The primary aim of this analysis was to understand the association of the exhaustion state of CD8[+] T cells with the TCR clonality in a large dataset. Due to the sparseness of CD8[+] T cells in the reference melanoma scRNA-seq data, the CD8[+] T-cell exhaustion trajectory was annotated using the exhaustion score[30] rather than any of the trajectory inference methods and validated by the expression pattern of the marker genes (Fig. 5b). Quantifying the exhaustion scores with two other independent gene sets gave rise to similar results, confirming the robustness of such an annotation (Supplementary Fig. 17).

We grouped 430 TCGA melanoma patients into tertiles according to the TCR expansion levels evaluated by MiXCR[31]. The MeDuSA deconvolution result showed an enrichment of CD8[+] T cells at the exhaustion state, which increased with the TCR expansion level (permutation F-test, P < 1 × 10[−4], with the P value capped by the number of permutations; Fig. 5c). In the terminal exhaustion state (that is, time 3, 66–100% of the cell trajectory), the correlation between TCR expansion level and CD8[+] T-cell abundance was 0.55 (P = 0.0025) (Supplementary Fig. 18),

suggesting that the exhaustion state of CD8[+] T cells was positively associated with TCR expansion level in melanoma.

The second aim of this analysis was to investigate the clinical relevance of the exhausted CD8[+] T cells. We first sought to examine the association of the exhaustion-state abundance of CD8[+] T cells with patients' overall survival. At each tertile of the exhaustion-state trajectory (time 1, 0–33% of the pseudotime; time 2, 33–66% of the pseudotime; time 3, 66–100% of the pseudotime), we divided TCGA melanoma patients into low and high groups (median cut-off) based on the average abundance of CD8[+] T cells. The result showed that only the abundance of CD8[+] T cells in the terminal exhaustion state was significantly associated with survival (time 3, log-rank-test, Hazard Ratio (HR) = 2.12, P = 8.2 × 10[−7]; Fig. 5d). We next examined the association of exhaustion-state abundance of CD8[+] T cells with patients' response to immune-checkpoint blockade (ICB). We collected a melanoma bulk RNA-seq dataset from anti-programmed cell death protein 1 (anti-PD1) pretreatment tumor tissues of 70 patients with metastatic skin melanoma. The MeDuSA deconvolution result suggested that the abundance of CD8[+] T cells at the terminal exhaustion state (time 3) was higher in anti-PD1 responders than that in anti-PD1 progressors (P = 0.0069) (Fig. 5e). Collectively, our results suggest that the abundance of CD8[+] T cells at the high-exhaustion state was positively correlated with TCR expansion level in melanoma and associated with patient's overall survival and response to anti-PD1 ICB.

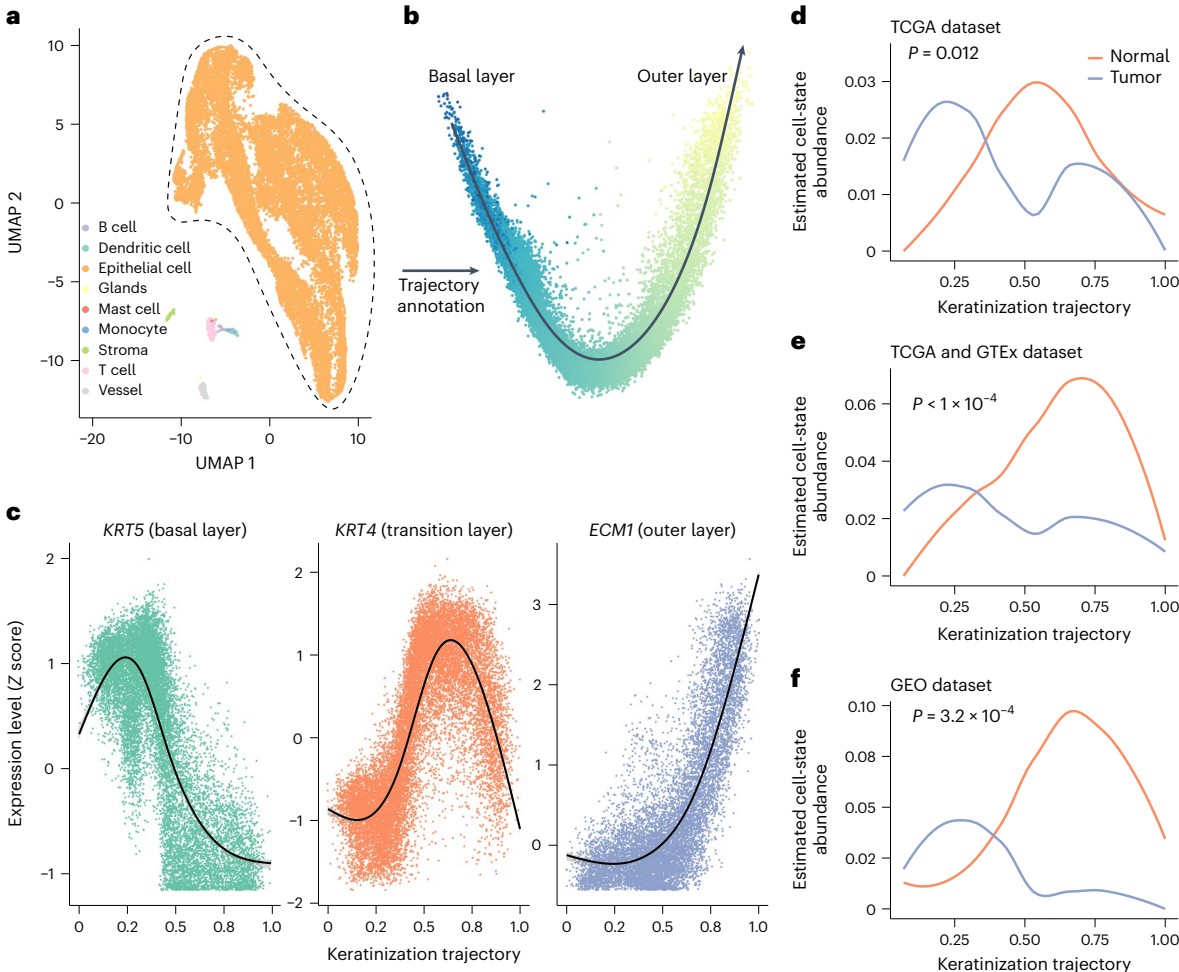

**Fig. 4 | Estimated epithelia abundance along the keratinization trajectory in normal and tumor esophagus tissues. a**, Uniform manifold approximation and projection (UMAP) embedding of the reference esophagus scRNA-seq data, where cells are colored according to their cell types (orange, epithelia). **b**, The keratinization trajectory of the epithelia in the reference scRNA-seq data. The black arrowed line represents the annotated trajectory using Slingshot, from the basal layer (germinative epithelium) to the outer layer (keratinized epithelium). **c**, The expression pattern of *KRT5* (marker gene of the basal layer), *KRT4* (marker gene of the transition layer) and *ECM1* (marker genes of the outer layer) confirmed the keratinization trajectory. The black lines represent the fitted curve using the LOESS and the shaded area indicates the 95% CI. **d–f**, The cell-state abundance of epithelia estimated by MeDuSA using a dataset from TCGA (**d**, $n = 109$), a combined set of data from TCGA and GTEx (**e**, $n = 664$) and a dataset from the Gene Expression Omnibus (GEO) (**f**, $n = 46$). Batch effects between the GTEx and TCGA datasets were adjusted using Combat-seq. The $x$ axis represents the keratinization trajectory, from the basal layer (left) to the outer layer (right). The curved line shows mean cell-state abundance across individuals. The $P$ values were computed using the permutation-based MANOVA-Pro method.

## Cell-state-dependent genetic regulation of gene expression

Finally, we applied MeDuSA to deconvolute cell-state abundances in an expression quantitative trait locus (eQTL) dataset (that is, a cohort with both single nucleotide polymorphism (SNP) genotype and bulk RNA-seq data) for detecting cell-state-dependent eQTLs (csd-eQTLs). Note that csd-eQTL mapping has been achieved only recently with cohort-level scRNA-seq data[32–35]. Using the esophagus scRNA-seq dataset above as the reference, we estimated the cell-state abundances along the epithelial differentiation trajectory in bulk RNA-seq data from 497 GTEx esophagus mucosa samples and computed the abundance of cells in each quartile of the epithelial differentiation trajectory for each sample (Fig. 6a). A csd-eQTL was claimed if the effect an SNP on bulk gene expression depended on cell-state abundance in any of the quartiles ('Mapping the cell-state-dependent eQTLs' in Methods). In total, we identified 162 genes with at least one csd-eQTL (defined as csd-eGenes) at 5% false-discovery rate (FDR) (Fig. 6b). The csd-eGenes were enriched in differentially expressed genes (DEGs) along the cell-state trajectory (fold enrichment = 2.12, 95% confidence interval (CI) 1.73–2.52; Fig. 6c,d), which could be replicated using the epithelial differentiation trajectory annotated by another independent esophagus scRNA-seq

dataset (Supplementary Fig. 19). We next annotated the epithelial differentiation trajectory using an independent esophagus single-cell assay for transposase-accessible chromatin (scATAC-seq) dataset (Fig. 6e–g) and tested for associations of the epithelial chromatin peaks with this trajectory ('Annotating the cell-state-dependent chromatin accessibility peaks' in Methods). We refer to the genomic regions with epithelial chromatin peaks associated with the differentiation trajectory (annotated by the scATAC-seq data) as cell-state-dependent open chromatin regions (csd-OCRs). We found that the lead csd-eQTLs were highly enriched (fold enrichment = 3.30, 95% CI 2.70–3.90) in the csd-OCRs, and the strength of enrichment increased with the significance level used to identify the csd-eQTLs (Fig. 6h). Taken together, we achieved csd-eQTL mapping in a conventional eQTL mapping dataset, and the identified csd-eQTLs were enriched in csd-OCRs and associated with genes enriched with cell-state specific expression (Fig. 6i).

## Discussion

In this study, we developed a cellular deconvolution method, MeDuSA, to estimate cell-state abundance over a one-dimensional trajectory in bulk RNA-seq data. Compared with other methods, the substantially

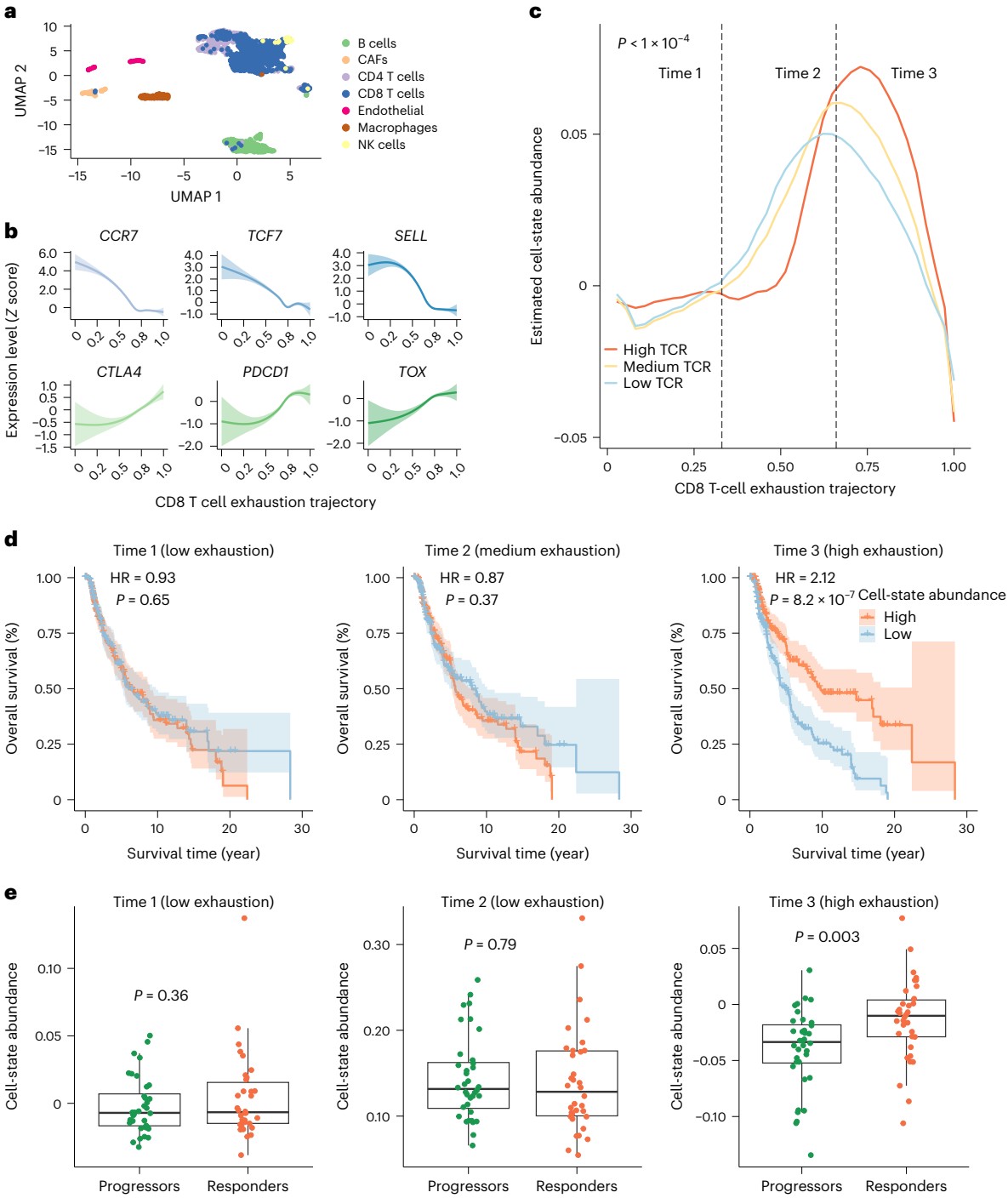

**Fig. 5 | Estimated abundance of CD8+ T cells along the exhaustion trajectory in skin melanoma. a**, UMAP embedding of melanoma reference scRNA-seq data, where cells are colored according to their cell types (blue, CD8+ T cells). CAFs, cancer-associated fibroblasts; NK cells, natural killer cells. **b**, Expression pattern of the marker genes along the exhaustion trajectory. The lines represent the fitted curve using the LOESS, and the shaded area indicates the 95% CI. **c**, Estimated cell-state abundances of CD8+ T cells in patients stratified into tertiles by TCR expansion level in TCGA data ($n = 430$). The x axis represents the exhaustion trajectory, from the naive state (left) to the exhausted state (right). The curved line shows mean estimated cell-state abundance across individuals. The P value was computed using MANOVA-Pro and was capped by the number of permutations. **d**, Kaplan–Meier plot for overall survival between two groups of patients with melanoma stratified by the estimated abundance of CD8+ T cells in

each tertile of the exhaustion state in TCGA data. The exhaustion-state tertiles are: time 1, the low-exhaustion state (0–33% of the exhaustion trajectory); time 2, the medium-exhaustion state (33–66% of the exhaustion trajectory); time 3, the high-exhaustion state (66–100% of the exhaustion trajectory). The shaded area represents the 95% CI of the fitted Kaplan–Meier curves. The P values were computed using a two-sided long-rank test. **e**, Boxplot of estimated cell-state abundance of CD8+ T cells in the patients with melanoma who did not respond to anti-PD-1 ICB ($n = 36$) versus the responders ($n = 34$) in each of the exhaustion-state tertiles. Each point represents one patient, color-coded based on their response to ICB. The P value was computed using a two-sided Wilcoxon test. The box indicates the IQR, the line within the box represents the median value and the whiskers extend to data points within 1.5 times the IQR.

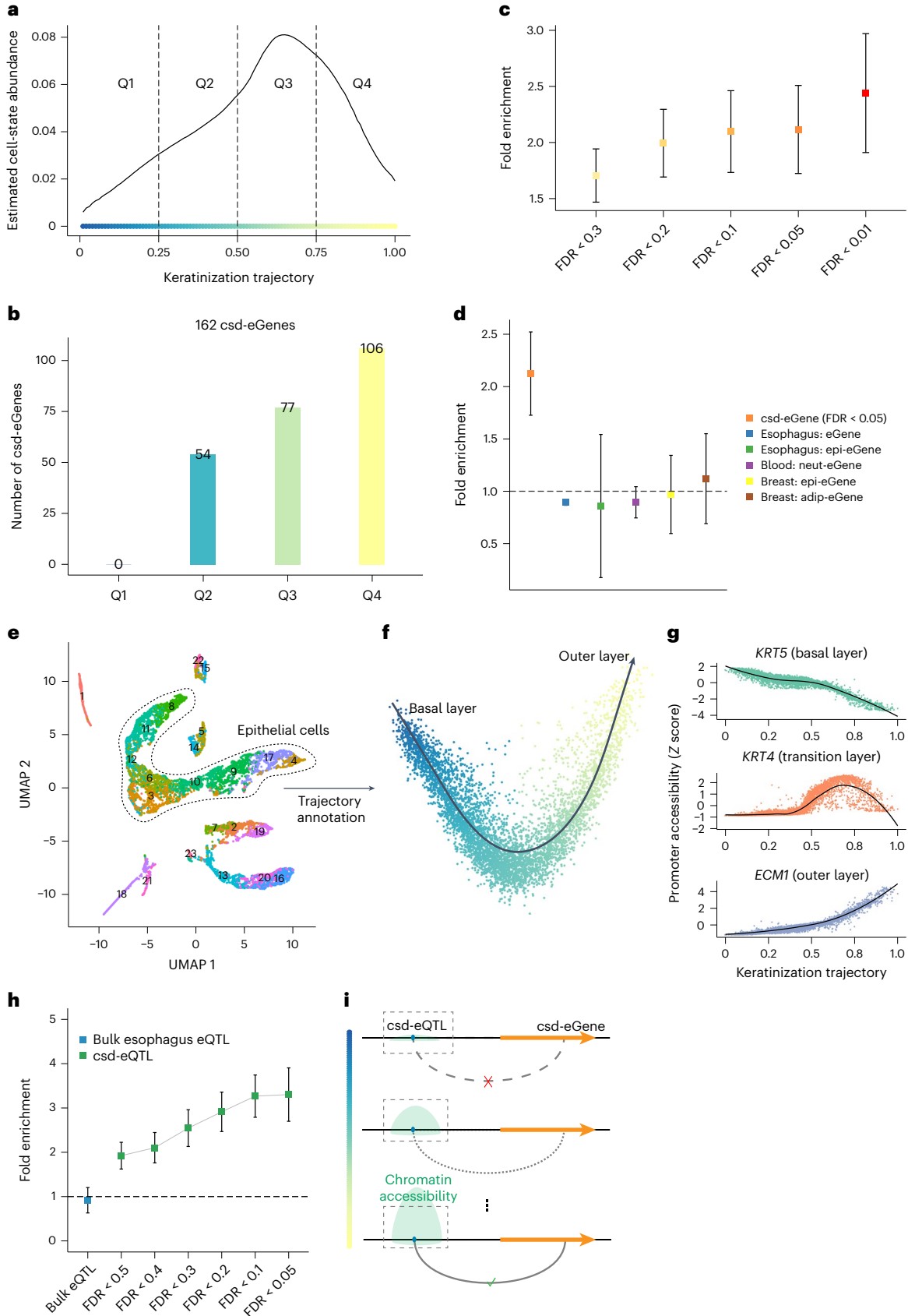

increased deconvolution accuracy of MeDuSA is mainly because of fitting the cells at the focal state as a fixed effect and the remaining cells individually as random effects. On average across the RNA-seq datasets used in this study, this approach explains an additional 10–40% of variance in bulk gene expression compared with the binning strategy (Supplementary Figs. 21 and 22).

**Fig. 6 | Identifying csd-eQTLs. a**, We stratified the estimated epithelia abundance along the differentiation (keratinization) trajectory into four quartiles (Q1–Q4). We then constructed a linear model with a cell-state-by-genotype interaction term ('Mapping the cell-state-dependent eQTLs' in Methods) to identify the cell-state (epithelial differentiation)-dependent eQTLs in the GTEx esophagus mucosa data ($n = 497$). **b**, The number of identified csd-eGenes in each quartile of the differentiation trajectory. Each column represents the number of csd-eGenes at different cell-state quartiles. **c**, The enrichment of the csd-eGenes in the cell-state trajectory DEGs, at different csd-eQTL FDR thresholds. Each data point indicates the estimated fold enrichment, color-coded according to the corresponding FDR thresholds as displayed on the *x* axis. The error bar represents the 95% CI computed using permutations ('Enrichment of eGenes in DEGs' in Methods). **d**, The enrichment of the csd-eGenes (csd-eQTL FDR < 0.05), eGenes (obtained from the GTEx eQTL data with FDR < 0.05) or cell-type-dependent eGenes (obtained from the GTEx cell-type-dependent eQTL data

with FDR < 0.05) in the cell-state trajectory DEGs. Each data point represents the estimated fold enrichment of eGenes, color-coded based on the corresponding tissue or cell type (epi, epithelial cells; neut, neutrophils; adip, adipose cells), with the error bars indicating the 95% CI of the estimated fold enrichment. **e**, UMAP embedding of the esophagus scATAC-seq data ($n = 3$). **f**, The annotated epithelial differentiation trajectory in the scATAC-seq data. The black arrowed line represents the annotated trajectory using Slingshot, from the basal layer to the outer layer. **g**, The cellular distributions of promoter accessibilities of *ECM1* (marker gene of the basal layer), *KRT4* (marker gene of the transition layer) and *KRT5* (marker genes of the outer layer) along the epithelial differentiation trajectory. The black lines represent the fitted curve using the LOESS, and the shaded area indicates the 95% CI. **h**, The enrichment of the lead csd-eQTLs or eQTLs in the csd-OCRs. Each data point represents the estimated fold enrichment of lead eQTLs at an FDR threshold, with the error bars representing the 95% CI. **i**, Conceptual illustration of the enrichment of the csd-eQTLs in the csd-OCRs.

MeDuSA is well-suited for biological scenarios where the underlying mechanisms involve continuous transitions of cellular states, such as cell development, differentiation or degeneration. In four case studies covering a broad range of research domains, we discovered that cell-state abundance was associated with disease conditions, clinical outcomes, mechanisms of pathogenicity and treatment exposures. These results recapitulated changes in cellular functions under different biological conditions, facilitating our understanding of cellular roles in disease etiology. Further, we showed that MeDuSA can be used to detect csd-eQTLs in bulk RNA-seq data. These results inform future studies to map csd-eQTLs in large cohorts and integrate the csd-eQTLs with data from genome-wide association studies to identify disease-relevant cell states and reveal the biological mechanisms underlying genetic associations for complex traits and diseases.

There are several caveats when applying MeDuSA in practice. First, the cell-state trajectory in reference scRNA-seq data needs to be pre-annotated. Although we have used different methods, including the diffusion map-based method (Slingshot), the RNA velocity-based method (scVelo) and the score-based method (CytoTRACE), for cell-trajectory inference, showing the compatibility of MeDuSA, a biased cell-state-trajectory annotation might result in biased cell-state abundance estimation. Second, the sequencing technology used to generate reference scRNA-seq data is another source of bias for deconvolution analyses. One of the greatest sources of bias in scRNA-seq is dropout events[36–38], especially for short-length methods such as those implemented by 10X Genomics. We corrected for this potential bias by filtering out genes expressed in less than 10% cells and averaging gene expression profiles of cells in the focal cell-state (Methods). It is of note that we have covered most common scRNA-seq platforms in simulations and applications including 10X Genomics, Drop-Seq, Seq-Well, C1, inDrop and Smart-seq2 (Supplementary Table 1). Imputing the reference scRNA-seq data by SAVER[39] did not improve the performance of MeDuSA in real-data applications (Extended Data Fig. 5). Third, the cell-state trajectory modeled in the current version of MeDuSA is a one-dimensional vector, which may not fully portray the complexity of cellular transitions, particularly in cases of multiple cell trajectories[40]. More work is warranted in the future to extend MeDuSA to model cell states on a multi-dimensional space. Fourth, a growing number of spatial transcriptomics studies have shown that cellular heterogeneity at spatial coordinates might be associated with unknown biological mechanisms[41–43]. In this regard, recovering spatial structures of bulk tissues using spatial transcriptomics data as a reference will be another interesting future direction to extend MeDuSA.

## Methods
### Ethical approval
This study was approved by the Ethics Committee of Westlake University (approval no. 20200722YJ001).

## The MeDuSA model
For a cell type of interest (that is, the focal cell type) in a tissue or cell line, the relative abundances of cells at different states (that is, cell-state abundances) can be estimated using a cell-trajectory analysis with scRNA-seq data. For a sample without scRNA-seq but with bulk RNA-seq data available, cell-state abundance can be estimated by projecting the RNA-seq data onto the cell-state trajectory derived from a reference scRNA-seq dataset[20]. More specifically, cells of the focal cell type in the reference scRNA-seq data are ranked by the cell-state trajectory, and a cell state is defined as a window on this trajectory. The window size can be customized, varying from a single cell to multiple neighboring cells at similar states. Given a specific window size, the cell trajectory in the reference can be subdivided into $m$ consecutive states. When the $i$th state is regarded as the focal state, the abundance of this state in the bulk RNA-seq data can, in principle, be estimated by the following model: $\mathbf{y} = \mathbf{x}_i\beta_i + \mathbf{e}$, where $\mathbf{y}$ is an $n \times 1$ vector comprising expression levels of a list of $n$ signature genes (selected to be associated with cell-state trajectory; section 1 of the Supplementary Note and Supplementary Fig. 30) in the bulk RNA-seq data, $\mathbf{x}_i$ is an $n \times 1$ vector comprising expression levels of the signature genes in cells at the focal state in the reference, with $\beta_i$ being the cell-state abundance to be estimated, and $\mathbf{e}$ is an $n \times 1$ vector of residuals, with $\mathbf{e} \sim N(\mathbf{0}, \mathbf{I}\sigma_e^2)$. If there are multiple cells at the focal state, the expression level of each gene is averaged across the cells.

A critical limitation of the above model is that the variance in $\mathbf{y}$ explained by $\mathbf{x}_i$ is likely to be minor, leaving a sizable residual variance and thereby considerable uncertainty in the estimated cell-state abundance $\hat{\beta}_i$. We propose to reduce the residual variance by fitting the focal state, together with the remaining cells of the focal cell type and the other cell types in the following LMM:

$$\mathbf{y} = \mathbf{x}_i\beta_i + \mathbf{C}\gamma + \mathbf{Z}\alpha + \mathbf{e} \tag{1}$$

where $\mathbf{y}$, $\mathbf{x}_i$, $\beta_i$ and $\mathbf{e}$ have the same definitions as above; $\mathbf{C}$ is matrix of gene expression levels, with each row representing a signature gene and each column representing the mean of each of the other cell types, and $\gamma$ is a vector of the corresponding effects; $\mathbf{Z}$ is also a matrix of gene expression levels, with each row representing a signature gene and each column representing each of the remaining cells of the focal cell type, and $\alpha$ is a vector of the corresponding effects. In this model, $\beta_i$ and $\gamma$ are treated as fixed effects, whereas $\alpha$ are treated as random effects, with $\alpha \sim N(\mathbf{0}, \boldsymbol{\Sigma})$, because the size of $\alpha$ ($k$ cells) is often larger than the size of $\mathbf{y}$ ($n$ signature genes). Under this model parameterization, we have $\mathbf{y} \sim N(\mathbf{x}_i\beta_i + \mathbf{C}\gamma, \mathbf{Z}\boldsymbol{\Sigma}\mathbf{Z}' + \mathbf{I}\sigma_e^2)$.

Compared with the strategy of binning cells by cell-state trajectory and fitting the mean of each bin in a regression model[1,2], this LMM has two distinct advantages. As cells, even those of the same type, are biologically heterogeneous, fitting the remaining cells of the focal cell type individually as random effects allows each cell to

have a specific weight on bulk gene expression, resulting in a better capturing of the variance in bulk gene expression and thereby a more precise estimate of the focal state in the fixed-effect term (that is, improved deconvolution accuracy). Second, the LMM ameliorates the collinearity problem between cells at the focal state (fitted as a fixed effect) and those at adjacent states (fitted as random effects) because of the shrinkage of random effects. For the other cell types, we fit the mean expression level of a whole cell type as a fixed-effect covariate rather than fitting individual cells as random effects for two reasons. First, the signature genes are selected to be associated with the cell-state trajectory in the focal cell type so that the associations of the signature genes with the other cell types are often weak. Second, fitting multiple random-effect components with near-zero variance often causes convergence problems in estimating the variance components.

In many LMM applications, random effects are assumed to be independent and identically distributed. However, the abundances of cells at adjacent states are likely to be correlated. To accommodate such correlations, we follow the previous work[44–46] to model the relationship between the abundance of cell $i$ (strictly speaking, the abundance of cells at a state represented by cell $i$) and those of the other cells as

$$\alpha_i = \theta \sum_{j=1, j \neq i}^{k} w_{ij} \alpha_j + \epsilon_i \qquad (2)$$

where $\alpha_i$ is the abundance of cell $i$, $\theta$ is a scaling factor, $w_{ij}$ is the weight between cells $i$ and $j$, and $\varepsilon_i$ is an error term with $\epsilon_i \sim N\left(0, \sigma^2_{\epsilon(i)}\right)$. Because cells at closer states tend to have higher correlations, we model the weight between cells $i$ and $j$ as $w_{ij} = \exp(-d_{ij}^2)$, with $d_{ij}$ being the Euclidian distance between the states on the cell-state trajectory[46,47]. Let $\mathbf{W} = \{w_{ij}\}$ be a $k \times k$ symmetric zero-diagonal matrix for all cell pairs and $\mathbf{D}$ be a diagonal matrix, with each diagonal element being the corresponding row sum of $\mathbf{W}$. We divide each $w_{ij}$ by $D_{ii}$ so that the sum of each row of $\mathbf{W}$ is unity. To ensure var($\alpha$) to be symmetric, we set $\sigma^2_{\epsilon(i)} = \lambda^2/D_{ii}$ with $\lambda$ being a scalar[48]. Following the Brook's factorization[45,49], we have var($\boldsymbol{\alpha}$) = $(\mathbf{D} - \theta\mathbf{W})^{-1}\lambda^2$. The distribution of $\mathbf{y}$ then becomes

$$\mathbf{y} \sim N(\mathbf{x}_i\beta_i + \mathbf{C}\gamma, \mathbf{V}) \qquad (3)$$

where $\mathbf{V} = \mathbf{Z}(\mathbf{D} - \theta\mathbf{W})^{-1}\lambda^2\mathbf{Z}' + \mathbf{I}\sigma_e^2$. We can fit this model iteratively for $i$ from 1 to $m$ to estimate $\beta_i$ for each focal state. Details of the derivation and parameter estimation of equation (3) are provided in the section 2 of the Supplementary Note. It should be noted the $\beta_i$ parameters represent the fractional abundances of different cell states within the focal cell type, which are bound between 0 and 1 and sum up to unity. To ensure unbiased estimation, the estimates of the $\beta_i$ parameters from the MeDuSA models are not constrained. However, for ease of interpretation, one can rescale the raw estimates to range from 0 to 1 and sum up to unity.

## Computational speed-up
Running the whole process above is time-consuming, largely because of the rate-limiting step of estimating $\mathbf{V}$ (strictly speaking, estimating the parameters to compute $\hat{\mathbf{V}}$), which needs to be done repeatedly for each focal state. Considering the minimal contribution of a focal cell state to the bulk gene expression level, we speed up the process by estimating $\mathbf{V}$ only once under the null model (that is, dropping the focal state from the fixed-effect terms and fitting all cells of the focal cell type in the random-effect term) and plug it in the generalized least squares[50] equation to compute $\hat{\beta}_i$ for each of the alternative models. This approximation has been widely used in LMM-based genetic association test[51–55], and has been proved to be accurate in our benchmark analyses (Supplementary Figs. 27 and 28).

## Smoothing
After estimating the cell-state abundances from the process above, we smooth the estimates over the cell-state trajectory by the locally estimated scatterplot smoothing (LOESS) or averaging the nearest neighbors. This smoothing process often leads to improved deconvolution accuracy due to reduced sampling variance of the estimates using the neighboring information.

## Simulation strategy
To make the simulation as close to reality as possible, we performed simulations using 17 real scRNA-seq datasets from different sequencing platforms and species. Each dataset was randomly split into two portions, one as the simulation source data and the other as the deconvolution reference data. The synthetic bulk RNA-seq data were generated as mixtures of scRNA-seq profiles based on the simulation source data. We grouped cells into $L$ uniformly distributed states and assigned an abundance ($a_l$) to each state ($l$) according to the pre-designed cell abundance distribution over the cell-state trajectory. To mimic the sampling variances in real bulk RNA-seq data, we randomly selected a certain number of cells (with replacement) from each state based on the assigned cell abundance. The pseudo bulk expression level was obtained by averaging the expression profiles of the selected individual cells. Specifically, the expression level of gene $g$ in the pseudo bulk RNA-seq data was generated as:

$$g = \frac{\sum_{l=1}^{L} \sum_{i=1}^{na_l} X_{g_i}^l}{n}$$

where $X_{g_i}^l$ is the expression level of gene $g$ of cell $i$ randomly selected from state $l$, and $n$ is the total number of selected cells. We set $n$ as $\min\{na_l \geq 1 | a_l \neq 0\}$ to ensure at least one selected cell for the non-empty states and rounded $na_l$ to an integer number. The cell-state abundance ($a_l$) was generated as a nonlinear function of the cell trajectory: $a_l = f(t_l)$ with $f$ being the shape mapping function and $t_l$ being the median trajectory rank of cells at state $l$. We designed four cell-state abundance distributions including:

monotonically increasing: $f(t) = t^k$
monotonically decreasing: $f(t) = (-t + 1)^k$

unimodal: $f(t) = \left[-(t - 0.5)^2 + \max\left((t - 0.5)^2\right)\right]^k$

bimodal: $f(t) = \sin(3\pi t) - \min(\sin(3\pi t))$

with $k$ being the curvature of the distribution. The generated cell-state abundances were then normalized so that they sum to unity. To further mimic differences in batch effects between scRNA-seq and bulk RNA-seq data, we added log-normally distributed noises to the pseudo bulk RNA-seq data[12]. The performances of MeDuSA and other methods under different levels of noises were shown in Supplementary Fig. 29.

## Testing for differences in cell-state abundances among groups
We propose an approach, MANOVA-Pro, that combines multiple analysis of variance (MANOVA) with polynomial regression to detect differences in cell-state abundance among groups (for example, case group versus control group). We first utilize the polynomial regression to model the distribution of cell-state abundance along the cell-state trajectory as

$$\boldsymbol{\beta}_j = \mathbf{T}\mathbf{b}_j + \mathbf{e} \qquad (5)$$

where $\beta_j$ is an $m \times 1$ vector of the estimated cell-state abundances of individual $j$ with $m$ being the number of states along the cell-state trajectory; $\mathbf{T} = [\mathbf{t}^0 \vdots \mathbf{t}^1 \cdots \mathbf{t}^{k-1}]$ is an $m \times k$ polynomial matrix with $\mathbf{t}$ being an $m \times 1$ cell-state vector and $(k - 1)$ being the polynomial degree; $\mathbf{b}_j$ is a $k \times 1$ vector of the regression coefficients corresponding to $\mathbf{T}$; $\mathbf{e}$ is a vector of the residuals, $\mathbf{e} \sim N(0, \mathbf{I}\sigma_e^2)$. The polynomial regression coefficients can be estimated as $\mathbf{b}_j = (\mathbf{T}'\mathbf{T})^{-1}\mathbf{T}'\beta_j$. We next perform an

MANOVA analysis to test if there is a difference in any of the polynomial regression coefficients among groups (for example, case versus control) based on the following model:

$$\mathbf{R} = \mathbf{H} + \mathbf{E} \qquad (6)$$

where $\mathbf{R} = \sum_{j=1}^{g} \sum_{i=1}^{n_j} (\mathbf{b}_{ji} - \bar{\mathbf{b}})(\mathbf{b}_{ji} - \bar{\mathbf{b}})^{\mathsf{T}}$ is a $k \times k$ matrix with $g$ being the number of groups, $\mathbf{b}_{ji}$ being a $k \times 1$ vector of the regression coefficients of individual $i$ in the group $j$, and $n_j$ being the number of individuals in the group $j$; $\mathbf{H} = \sum_{j=1}^{g} n_j (\bar{\mathbf{b}}_{j.} - \bar{\mathbf{b}})(\bar{\mathbf{b}}_{j.} - \bar{\mathbf{b}})^{\mathsf{T}}$ is the hypothesis sum of squares and cross products matrix; $\mathbf{E} = \sum_{j=1}^{g} \sum_{i=1}^{n_j} (\mathbf{b}_{ji} - \bar{\mathbf{b}}_{j.})(\mathbf{b}_{ji} - \bar{\mathbf{b}}_{j.})^{\mathsf{T}}$ is the error sum of squares and cross products matrix. We can use the following $F$ statistic to test against the null hypothesis $H_0 : \mathbf{b}_1 = \mathbf{b}_2 = \cdots = \mathbf{b}_g$,

$$F = \frac{\Lambda(2u + s + 1)}{(s - \Lambda)(2m + s + 1)} \qquad (7)$$

where $\Lambda$ is the Pillai trace with $\Lambda = \mathrm{tr}\left(\mathbf{H}(\mathbf{H} + \mathbf{E})^{-1}\right)$, $s = \min(g - 1, k)$, $m = (|k - (g - 1)| - 1)/2$ and $u = \left(\sum_{j=1}^{g} n_j - k - g - 1\right)/2$. Under the null hypothesis, this $F$ statistic follows an $F$ distribution with $s(2m + s + 1)$ and $s(2u + s + 1)$ degrees of freedom.

### Correcting for inflation in association test

An important application of the estimated cell-state abundance is to test its association with a phenotype, for example, case-control status, across individuals. Such an analysis can be performed using the MANOVA-PRo method above that tests the association of cell-state abundance with a categorical phenotype, accounting for the relationship between the cell-state abundance and cell trajectory. Alternatively, if the interest is to test whether the abundance of cells in a specific state (or a bin of states without concerning the relationship between the cell-state abundance and cell trajectory within the bin), then the association test can be performed under the linear regression model framework. However, we have observed from simulations that all the association tests mentioned above can suffer from inflation because the estimated cell-state abundance is correlated across individuals, owing to the correlation of gene expression, and such correlation can be group dependent. For example, we observed in multiple bulk RNA-seq datasets that the mean correlation of gene expression was higher within the case or control group than that between groups (Supplementary Fig. 23). One extreme example was that the difference in estimated cell-state abundance between the case and control groups was statistically significant even if the reference scRNA-seq data were randomly generated (Supplementary Fig. 24). Such inflation also probably exists in cell-type deconvolution analyses, as demonstrated in our simulations (Supplementary Fig. 25). To account for such correlation-induced inflation, we propose to assess the significance level of the association by permutation test. In each permutation, we randomly shuffle the signature genes, and re-run the cellular deconvolution analysis and the subsequent association analysis. We repeat the permutation 1,000 times (or 10,000 times when necessary) and compute an empirical $P$ value by comparing the observed association test statistic with the test statistics obtained from permutations. We have demonstrated by simulations under various conditions that the permutation-based test was well calibrated under the null of no association (Supplementary Fig. 26).

### Processing the scRNA-seq, bulk RNA-seq and scATAC-seq data

We used 24 scRNA-seq datasets from the public domain (see Supplementary Table 1 for the information about species, sample size, sequencing platform and data access). Among them, 17 scRNA-seq datasets were used in simulation analyses, with the cell-state trajectory annotated previously or in this study using CytoTRACE[56]. In addition, we used 21 bulk RNA-seq datasets, with relevant tissue, sample size and data access information compiled in Supplementary Table 2. We also utilized scATAC-seq data from three human esophagus samples. The procedures for processing the scRNA-seq, bulk RNA-seq and scATAC-seq data are details in sections 4–6 of the Supplementary Note.

### Mapping the cell-state-dependent eQTLs

We used SNP genotype data of 497 GTEx samples. Following the GTEx pipeline, we performed a standard quality control process of the genotype data using PLINK2[57], with the parameters '−geno 0.01−maf 0.05− hwe 0.000001−mind 0.01'. The workflow for mapping the cell-state-dependent eQTLs (csd-eQTLs) is illustrated in Supplementary Fig. 35. The csd-eQTLs were mapped using a linear regression model with an interaction term between SNP genotype and the estimated cell-state abundance: $y_i = x_i\alpha + s_i\beta + x_i s_i\gamma + \sum_j c_{ij}\delta_j + e_i$, where $y_i$ is the gene expression level of the $i$th individual, $x_i$ is the genotype variable of an SNP, $s_i$ is the overall abundance of cells at a range of states (for example, one of the quartiles of the cell-state trajectory), $x_i s_i$ is the interaction term, $c_{ij}$ represents the $j$th covariate, and $e_i$ is the residual. Following the standard eQTL mapping pipeline[58], we used age, sex, the top-five genotype principal components (to correct for population stratification), and 60 PEER[59] factors (to correct for biological/technical confounding factors) as the covariates. For each gene, only SNPs within the $cis$ window (that is, ±1 Mb) of the transcription start site were tested. To avoid outlier effects, we performed the rank-based inverse normal transformation of the TMM (i.e., trimmed mean of m-values) normalized gene expression values and the cell-state abundances. We filtered out SNPs with minor allele frequency <0.05. For each of the SNPs retained, we tested the significance of the interaction term for csd-eQTL detection. Following the pipeline of mapping cell-type-dependent eQTLs[60], we used eigenMT[61] to correct for multiple testing in each $cis$ window. We then computed the Benjamin–Hochberg FDR values based on the eigenMT adjusted $P$ values to determine the experimental-wise significance threshold. The above csd-eQTL mapping process was conducted using the software tensorQTL[62].

### Annotating the cell-state-dependent chromatin accessibility peaks

We performed dimension-reduction analysis for epithelia in the scATAC-seq data using the same pipeline described above. We used Slingshot[24] to annotate the epithelial keratinization (differentiation) trajectory based on the top two eigenvectors (Supplementary Fig. 20). To avoid potential outlier effects, we eliminated chromatin accessibility peaks that were available in less than 5% of the epithelial cells. For each of the remaining peaks, we utilized the generalized additive model implemented in the R package 'mgcv' to identify epithelial differentiation-dependent accessible chromatin peaks: $\mathbf{y} \approx \mathbf{s}(\mathbf{x}) + \mathbf{C}$, where $\mathbf{y}$ is a vector of chromatin accessibility peaks across cells, $\mathbf{x}$ is a vector of pseudotime values of the epithelial keratinization trajectory, $\mathbf{s}$ is the smoothing spline representing the linear combination of cubic basis functions and $\mathbf{C}$ is the matrix of covariates. We added the total number of fragments of each cell to account for the variation in sequencing depth[63]. We used the total number of fragments and donor of the cell as covariates. Following previous studies[63,64], we assumed that the chromatin accessibility peaks follow a negative binomial distribution. The strength of association between chromatin accessibility and epithelial differentiation trajectory was quantified by the $\chi^2$ value of the smoothing spline.

### Enrichment of eQTLs for chromatin accessibility

We assigned cell-state-dependent chromatin accessibility $\chi^2$ values obtained above to the SNPs included in the csd-eQTL analysis. SNPs located in regions without chromatin accessibility data were excluded. To avoid ascertainment bias, we randomly sampled control SNPs from null SNPs, ensuring that their number and minor allele frequency distribution matched with those of the SNPs in query. The sampling process was repeated 1,000 times. The fold enrichment was calculated

by dividing the mean $\chi^2$ value of the SNPs in query by the mean of mean $\chi^2$ values across the 1,000 sets of control SNPs. We employed the delta method[65,66] to compute the sampling variance of the fold enrichment. Specifically, let $x$ be the mean $\chi^2$ value of the SNPs in query and $y = \{y_1, y_2, \ldots, y_i, \ldots, y_m\}$ with $y_i$ being the mean $\chi^2$ value of $i$th set of control SNPs. The fold enrichment was estimate as $x/\bar{y}$, with $\bar{y}$ being the mean across $m$ replicates ($m = 1,000$ in this case). The variance of $x/\bar{y}$ is expressed as: $\mathrm{var}\left(\frac{x}{\bar{y}}\right) = \left(\frac{x}{\bar{y}}\right)^2 \left[\frac{\mathrm{var}(x)}{x^2} + \frac{\mathrm{var}(\bar{y})}{\bar{y}^2} - \frac{2\mathrm{cov}(x,\bar{y})}{x\bar{y}}\right]$. Assuming that $\mathrm{cov}(x,\bar{y}) \approx 0$, and $\mathrm{var}(x) \approx \widehat{\mathrm{var}}(y)$, with $\widehat{\mathrm{var}}(y)$ being the observed variance of $y$ across replicates, the sampling variance of the fold enrichment estimate can be computed as: $\mathrm{var}\left(\frac{x}{\bar{y}}\right) \approx \left(\frac{x}{\bar{y}}\right)^2 \left[\frac{\widehat{\mathrm{var}}(y)}{x^2} + \frac{\widehat{\mathrm{var}}(y)}{m\bar{y}^2}\right]$.

### Enrichment of eGenes in DEGs

We allocated the $\chi^2$ values of the DEGs to the genes involved in the csd-eQTL analysis. Employing a similar method as previously mentioned, we computed the fold enrichment of eGenes by dividing the average $\chi^2$ value of the eGenes in query by the mean of mean $\chi^2$ values obtained from 1,000 sets of randomly chosen control genes. The delta method was used to calculate the sampling variance of the fold enrichment.

### Statistics and reproducibility

The $P$ values to test for differences in cell-state abundances among groups were calculated using the permutation-based MANOVA-Pro method. For the survival analysis, $P$ values were computed using a two-sided log-rank test. Csd-eQTLs $P$ values were computed using a one-sided chi-squared test. Enrichment $P$ values for the csd-eGenes and csd-eQTLs were derived through permutations. The sample size for each analysis was determined by the maximum number of eligible samples available in the respective datasets. The study design did not require randomization or blinding. To reproduce the primary results of this research, refer to the analytical pipeline available at https://github.com/LeonSong1995/MeDuSA_Analysis.

### Reporting summary

Further information on research design is available in the Nature Portfolio Reporting Summary linked to this article.

## Data availability

All the scRNA-seq, scATAC-seq and bulk RNA-seq data used in this study are available in the public domain with the relevant information summarized in Supplementary Tables 1 and 2. The GTEx genotype data is available at https://gtexportal.org/home/protectedDataAccess. The GTEx eQTLs summary data is available at https://gtexportal.org/home/datasets. The csd-eQTLs summary data is available at https://doi.org/10.5281/zenodo.8018006 ref. 67. The GRCh38 genome is available at https://www.ncbi.nlm.nih.gov/projects/genome/guide/human. The GENECODE-v38 transcriptome reference is available at https://www.gencodegenes.org/human. Source data for Figs. 2–6 and Extended Data Figs. 1–5 are available with this paper.

## Code availability

The source code of MeDuSA is available at https://github.com/LeonSong1995/MeDuSA ref. 68.

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

## Acknowledgements

This research was supported by the Leading Innovative and Entrepreneur Team Introduction Program of Zhejiang (2021R01013), 'Pioneer' and 'Leading Goose' R&D Program of Zhejiang (2022SDXHDX0001), Research Program of Westlake Laboratory of Life Sciences and Biomedicine (202208013) and Research Center for industries of the Future (RCIF) at Westlake University. The funders had no role in study design, data collection and analysis, decision to publish or preparation of the manuscript. We thank F. Cheng and T. Xu for helpful discussions and the Westlake University High-Performance Computing Center for assistance in computing. This study used the data from the GTEx (dbGaP accession phs000178) and the TCGA (dbGaP accession phs000424).

## Author contributions

J.Y. and L.S. conceived the study. J.Y., L.S., T.Q. and X.S. designed the experiment. L.S. and J.Y. developed the methods with input from X.S. L.S. developed the software tool, curated the data and conducted all analyses with the assistance and guidance from J.Y., T.Q. and X.S. J.Y. supervised the project. L.S. and J.Y. wrote the paper with the participation of all authors. All authors reviewed and approved the final paper.

## Competing interests

The authors declare no competing interests.

## Additional information

**Correspondence and requests for materials** should be addressed to Jian Yang.

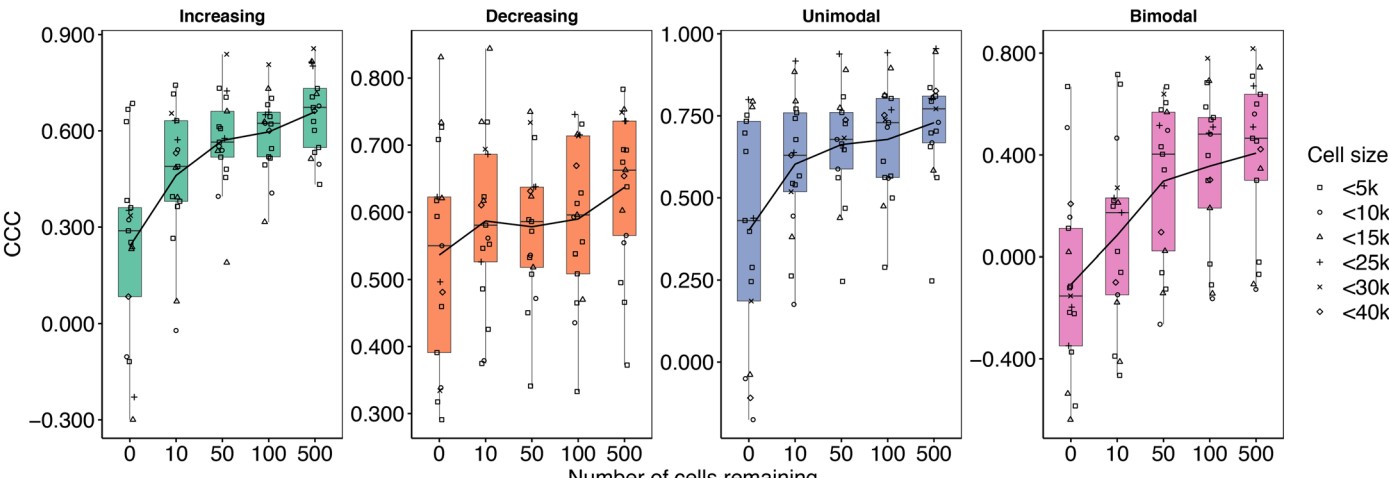

**Extended Data Fig. 1 | Deconvolution accuracy of MeDuSA-NS with decreasing number of cells fitted in the random-effect component.** We grouped cells of the focal cell type into ten uniformly distributed cell bins over the cell-state trajectory and randomly sampled a subset of cells from each cell bin to be fitted in the random-effect component of the MeDuSA-NS model. The x-axis is the number of cells fitted in the random-effect component. Each dot represents the mean deconvolution accuracy over five replicates for one simulation source data, colored by the number of cells in the data. The box indicates the interquartile IQR, the line within the box represents the median value, and the whiskers extend to data points within 1.5 times the IQR.

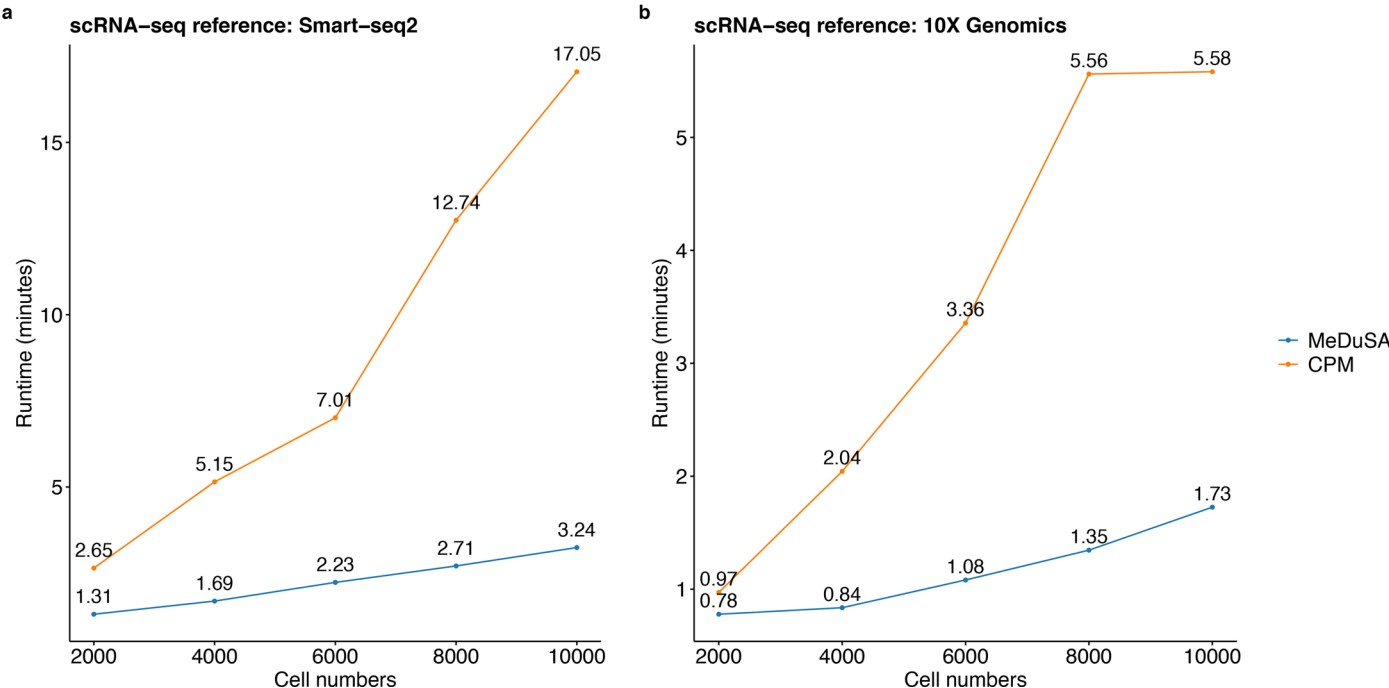

**Extended Data Fig. 2 | Runtime of MeDuSA and CPM.** Panel a and b shows the runtime of MeDuSA and CPM to deconvolute a bulk RNA-seq dataset using a Smart-seq2 or 10X Genomics scRNA-seq dataset, respectively, as the reference.

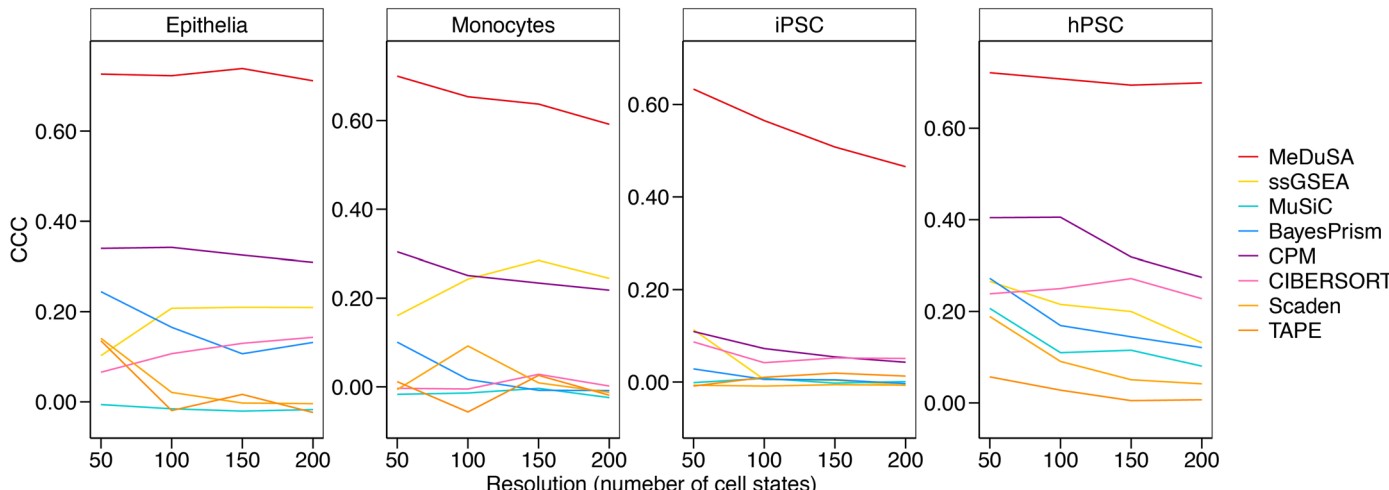

**Extended Data Fig. 3 | Deconvolution accuracy of MeDuSA and other methods with different resolutions in the real-data benchmark analysis.** The x-axis represents deconvolution resolution (as measured by the number of cell states), and the y-axis represents the deconvolution accuracy (as measured by CCC).

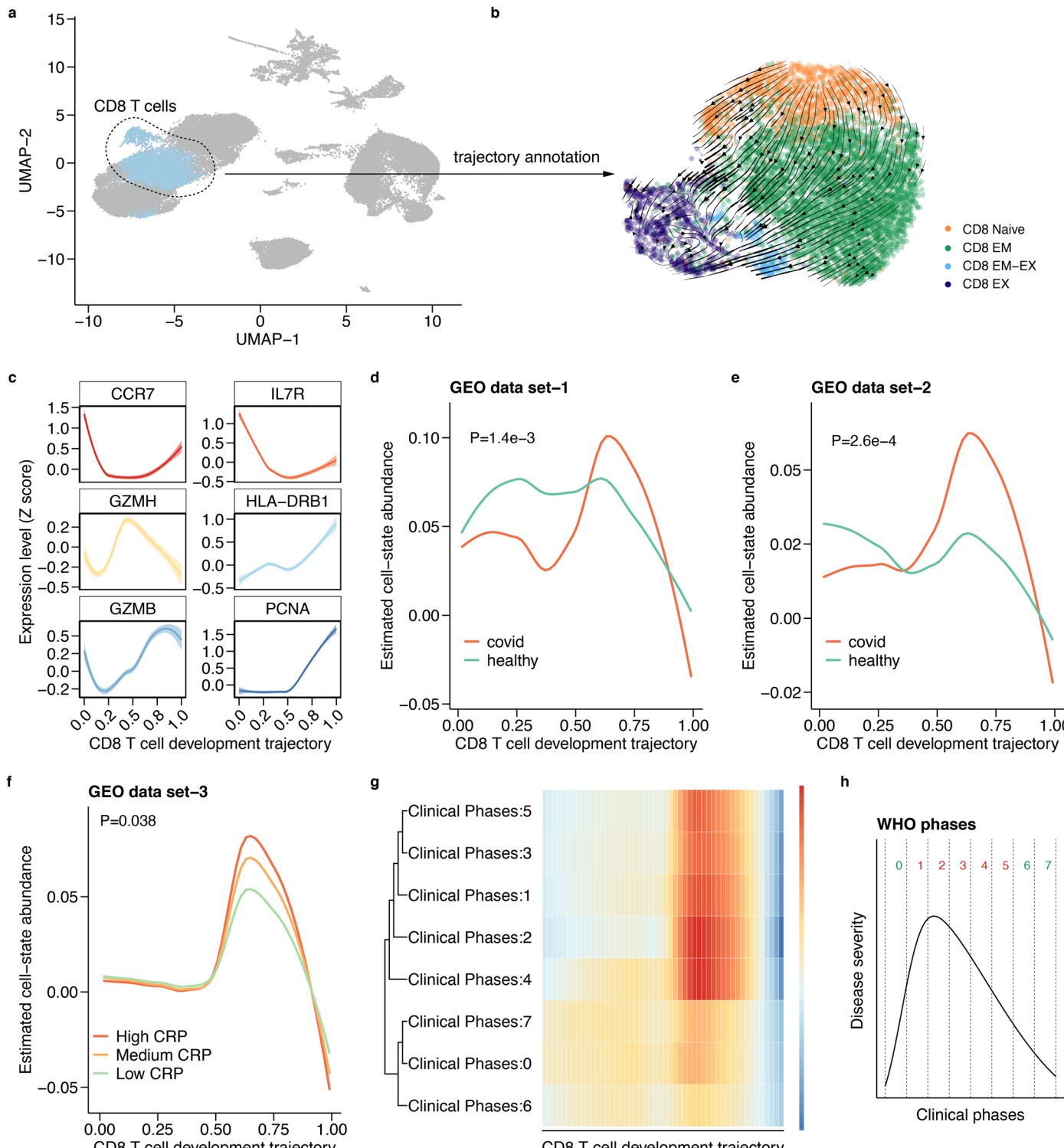

**Extended Data Fig. 4 | Estimated abundance of CD8+ T cells along the development trajectory in COVID-19.** (**a**) UMAP embedding of the reference covid-19 scRNA-seq data, where cells are colored according to their cell types (azure, CD8+ T cells). (**b**) RNA velocity analysis (scVelo) suggesting that CD8+ T cells developed from the naïve state (Tn) to the exhaustion state (Tex). Colors represent subtypes of CD8+ T cells (orange, naive CD8+ T cells; green, effective memory CD8+ T cells; blue, effective-exhaustion transition CD8+ T cells; purple, exhausted CD8+ T cells). (**c**) Profiling marker genes to confirm the development trajectory. The lines represent the fitted curve using the LOESS, and the shaded area indicates the 95% CI. (**d**) Estimated cell-state abundances of CD8+ T cells along the development trajectory from bulk RNA-seq data of COVID-19 patients

(n = 17) compared with those from healthy donors (n = 17). (**e**) Replicating the results presented in panel d in an independent bulk RNA-seq dataset (n = 54). (**f**) Estimated cell-state abundances of CD8+ T cells in COVID-19 patients stratified into tertiles by blood CRP level (n = 100). The x-axis represents the development trajectory, from the naïve state (left) to the exhausted state (right). The curved line shows mean cell-state abundance across individuals. The p values were computed using the permutation-based MANOVA-Pro method. (**g**) Heatmap of estimated cell-state abundances of CD8+ T cells in eight groups of COVID-19 patients stratified by the WHO clinical phase (n=27). (**h**) Conceptual illustration of the WHO clinical phase, reflecting disease severity during the SARS-CoV-2 infection.

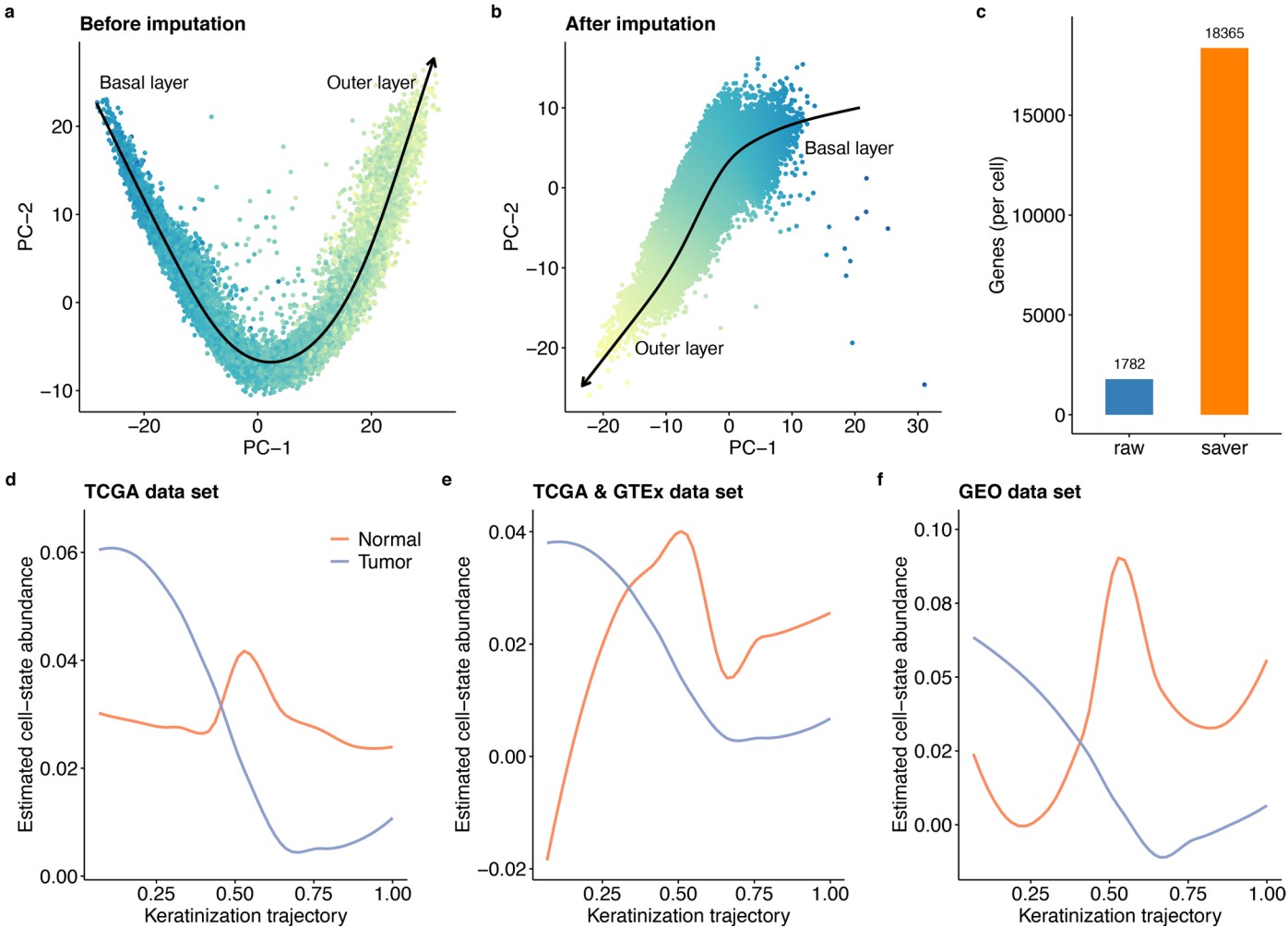

**Extended Data Fig. 5 | Estimated epithelial abundance along the keratinization trajectory in normal esophagi and tumors.** (**a**, **b**) PCA embedding of the reference scRNA-seq data before (panel a) and after (panel b) performing gene expression imputation using SAVER, where cells are colored according to their states. The black arrowed line represents the annotated trajectory using Slingshot, from the basal layer (germinative epithelium) to the outer layer (keratinized epithelium). (**c**) Number of genes expressed per cell before and after gene expression imputation. We used the SAVER imputed scRNA-seq data as the reference for the cell-state deconvolution analysis below. (**d**–**f**) Estimated cell-state abundances of epithelial cells using a dataset from TCGA data (n = 109), a combined set of data from TCGA and GTEx data (n = 664), and a dataset from the GEO (n = 46). The x-axis shows the keratinization trajectory, from the basal layer (left) to the upper layer (right). The curved line represents mean estimated cell-state abundances across individuals.

# Reporting Summary

## Statistics

For all statistical analyses, confirm that the following items are present in the figure legend, table legend, main text, or Methods section.

| n/a | Confirmed | |
|---|---|---|
| ☐ | ☒ | The exact sample size (*n*) for each experimental group/condition, given as a discrete number and unit of measurement |
| ☐ | ☒ | A statement on whether measurements were taken from distinct samples or whether the same sample was measured repeatedly |
| ☐ | ☒ | The statistical test(s) used AND whether they are one- or two-sided<br>*Only common tests should be described solely by name; describe more complex techniques in the Methods section.* |
| ☐ | ☒ | A description of all covariates tested |
| ☐ | ☒ | A description of any assumptions or corrections, such as tests of normality and adjustment for multiple comparisons |
| ☐ | ☒ | A full description of the statistical parameters including central tendency (e.g. means) or other basic estimates (e.g. regression coefficient) AND variation (e.g. standard deviation) or associated estimates of uncertainty (e.g. confidence intervals) |
| ☐ | ☒ | For null hypothesis testing, the test statistic (e.g. *F*, *t*, *r*) with confidence intervals, effect sizes, degrees of freedom and *P* value noted<br>*Give P values as exact values whenever suitable.* |
| ☒ | ☐ | For Bayesian analysis, information on the choice of priors and Markov chain Monte Carlo settings |
| ☒ | ☐ | For hierarchical and complex designs, identification of the appropriate level for tests and full reporting of outcomes |
| ☐ | ☒ | Estimates of effect sizes (e.g. Cohen's *d*, Pearson's *r*), indicating how they were calculated |

*Our web collection on statistics for biologists contains articles on many of the points above.*

## Software and code

Policy information about availability of computer code

| Data collection | No data-collection software was used. |
|---|---|
| Data analysis | We implemented our method in a R package called MeDuSA. The source code is freely available at https://github.com/LeonSong1995/MeDuSA. CellRanger v6.1.2, R v4.1.0 (package: Seurat v3, SingleR v1.8.0, Slingshot v2.8.0, SCORPIUS v1.0.8, TSCAN v1.38.0, SnapATAC v2.0, CytoTrace v0.3.3), and Python v3.8 (package: scVelo v0.2.5) were used for the quality control and statistical analyses of the  scRNA-seq and scATAC-seq data. MACS2 v2.2.7 was used for calling chromatin peaks of the scATAC-seq data. STAR v2.7.9 , RSEM v1.1.17 and Fastp v 0.23.2 were used for the quality control and statistical analyses of the bulk RNA-seq data. MIXCR v3.0.13 was used for the T cell receptor expansion analyses. R v4.1.0 (package: scBio v0.1.5, MuSiC v1.00, Cibersort v1.00, EPIC v1.16, GSVA v1.48.1, and BayesPrism v2.0) and Python v3.10 (package: Scaden v1.10 and TAPE v1.12) were used for the cell type (state) deconvolution analyses. Plink v2.0 was used for the quality control of the genotype data. TensorQTL v1.0.7 was used for the eQTL analysis. |

For manuscripts utilizing custom algorithms or software that are central to the research but not yet described in published literature, software must be made available to editors and reviewers. We strongly encourage code deposition in a community repository (e.g. GitHub). See the Nature Portfolio guidelines for submitting code & software for further information.

## Data

Policy information about <u>availability of data</u>

All manuscripts must include a <u>data availability statement</u>. This statement should provide the following information, where applicable:

- Accession codes, unique identifiers, or web links for publicly available datasets
- A description of any restrictions on data availability
- For clinical datasets or third party data, please ensure that the statement adheres to our <u>policy</u>

All the scRNA-seq, scATAC-seq, and bulk RNA-seq data used in this study are available in the public domain with the relevant information summarized in Supplementary Tables 1-2. The GTEx genotype data is available at: https://gtexportal.org/home/protectedDataAccess. The GTEx eQTLs summary data is available at: https://gtexportal.org/home/datasets. The csd-eQTLs summary data is available at zenodo: https://doi.org/10.5281/zenodo.8018006. The GRCh38 genome is available at: https://www.ncbi.nlm.nih.gov/projects/genome/guide/human. The GENECODE-v38 transcriptome reference is available at:https://www.gencodegenes.org/human. Source data for Figures 2-6 and Extended Data Figures 1-5 are available with this manuscript.

## Human research participants

Policy information about <u>studies involving human research participants and Sex and Gender in Research.</u>

| | |
|---|---|
| Reporting on sex and gender | We analyzed existing data sets and collected the age and gender from the corresponding public data domain. We have included age and gender as covariates in analysis. |
| Population characteristics | Our study involved publicly available datasets (e.g., GTEx, TCGA and existing summary statistics). We have included data from different ancestries. |
| Recruitment | We analyzed existing data sets. Thus, no recruitment was performed. |
| Ethics oversight | Ethics Committee of Westlake University (No. 20200722YJ001) |

Note that full information on the approval of the study protocol must also be provided in the manuscript.

# Field-specific reporting

Please select the one below that is the best fit for your research. If you are not sure, read the appropriate sections before making your selection.

☒ Life sciences  ☐ Behavioural & social sciences  ☐ Ecological, evolutionary & environmental sciences

For a reference copy of the document with all sections, see nature.com/documents/nr-reporting-summary-flat.pdf

# Life sciences study design

All studies must disclose on these points even when the disclosure is negative.

| | |
|---|---|
| Sample size | This study uses twenty-four scRNA-seq and twenty-one bulk RNA-seq datasets throughout the simulation, validation, and four applications. The sample sizes of all the datasets have been described in Supplementary Tales 1 and 2. In the simulation study, the cell number across seventeen scRNA-seq datasets totaled 185,012, determined by the maximum number of eligible cells after implementing quality control measures. For the validation study, we included 4,684 samples from twelve distinct tissues (cell lines), which were determined by the maximum number of eligible samples in the respective datasets. For the case studies, the sample size amounted to 711 for the esophagus data, 215 for the COVID-19 data, and 507 for the melanoma data, based on the maximum number of eligible samples in the corresponding datasets. For the csd-eQTL mapping, the sample size is 497, which was determined by the maximum number of unrelated individuals possessing both SNP genotype and esophagus bulk RNA-seq data in the GTEx dataset. |
| Data exclusions | For the scRNA-seq data, we excluded cells with >4500 and <2000 expressed genes (potential duplets or empty droplets). For the bulk RNA-seq data, we excluded individuals with ambiguous clinical diagnoses. |
| Replication | We repeated the simulation with each specific setting multiple times to assess the robustness of the deconvolution method. All the simulation replications were successfully performed. In the real data application for esophageal samples, we reproduced the results in three distinct bulk RNA-seq datasets and two scRNA-seq datasets. Regarding the real data application for COVID-19, we replicated the findings across three bulk RNA-seq datasets. For csd-eQTL mapping, we repeated the enrichment analysis using two distinct scRNA-seq datasets. The results demonstrated consistency across all replications. |
| Randomization | We performed analyses of the existing datasets; therefore, no randomization was implemented with regard to data generation. In the simulation analysis, we divided the scRNA-seq dataset into two sections at random. One section was then randomly designated as the simulation source data, and the other portion used as the reference data for deconvolution analysis. |
| Blinding | We performed analyses of the existing datasets, and as such,no blinding measures were implemented in this study. |

# Reporting for specific materials, systems and methods

We require information from authors about some types of materials, experimental systems and methods used in many studies. Here, indicate whether each material, system or method listed is relevant to your study. If you are not sure if a list item applies to your research, read the appropriate section before selecting a response.

## Materials & experimental systems

| n/a | Involved in the study |
|-----|----------------------|
| ☒ | ☐ Antibodies |
| ☒ | ☐ Eukaryotic cell lines |
| ☒ | ☐ Palaeontology and archaeology |
| ☒ | ☐ Animals and other organisms |
| ☒ | ☐ Clinical data |
| ☒ | ☐ Dual use research of concern |

## Methods

| n/a | Involved in the study |
|-----|----------------------|
| ☒ | ☐ ChIP-seq |
| ☒ | ☐ Flow cytometry |
| ☒ | ☐ MRI-based neuroimaging |

