## [Peer Review File · Nature Computational Science]

Peer Review Information

Journal: Nature Computational Science

Manuscript Title: Mixed model-based deconvolution of cell-state abundances (MeDuSA) along a one-dimensional trajectory

Corresponding author name(s): Professor Jian Yang

Reviewer Comments & Decisions:

Decision Letter, initial version:
--

Date: 25th January 23 15:33:32

Last Sent: 25th January 23 15:33:32

Triggered By: Fernando Chirigati

From: fernando.chirigati@us.nature.com

To: jian.yang@westlake.edu.cn

CC: jie.pan@us.nature.com

Subject: Decision on Nature Computational Science manuscript NATCOMPUTSCI-22-1257

Message: ** Please ensure you delete the link to your author homepage in this e-mail if you wish to forward it to your co-authors. **

Dear Professor Yang,

Your manuscript "MeDuSA: mixed model-based deconvolution of cell-state abundance" has now been seen by 3 referees, whose comments are appended below. You will see that while they find your work of interest, they have raised points that need to be addressed before we can make a decision on publication.

The referees' reports seem to be quite clear. Naturally, we will need you to address *all* of the points raised.

While we ask you to address all of the points raised, the following points need to be substantially worked on:

- Please improve your benchmark analysis as requested by Referee #3; real data samples across diverse tissue types with known cell-state abundances are needed.
- Please revise the manuscript thoroughly to avoid overstatements.
- Please add sufficient methodological details in the paper, as well as plots/data for the assessment of all of the experimental results.
- Please strengthen the motivation for this tool, as requested by Referee #3: why is

this tool needed when compared to simpler gene-based approaches? Please add new experimental data if needed.

- For the code, please (i) add more documentation to it; (ii) provide a more user-friendly pipeline for running the code; (iii) add instructions on how to best preprocess the expression data; and (iv) provide a mechanism to easily reproduce all of the experiments of the paper.

Please use the following link to submit your revised manuscript and a point-by-point response to the referees' comments (which should be in a separate document to any cover letter):

[REDACTED]

** This url links to your confidential homepage and associated information about manuscripts you may have submitted or be reviewing for us. If you wish to forward this e-mail to co-authors, please delete this link to your homepage first. **

To aid in the review process, we would appreciate it if you could also provide a copy of your manuscript files that indicates your revisions by making use of Track Changes or similar mark-up tools. Please also ensure that all correspondence is marked with your Nature Computational Science reference number in the subject line.

In addition, please make sure to upload a Word Document or LaTeX version of your text, to assist us in the editorial stage.

If you have any issues when updating your Code Ocean capsule during the revision process, please email the Code Ocean support team Cc'ing me.

To improve transparency in authorship, we request that all authors identified as 'corresponding author' on published papers create and link their Open Researcher and Contributor Identifier (ORCID) with their account on the Manuscript Tracking System (MTS), prior to acceptance. ORCID helps the scientific community achieve unambiguous attribution of all scholarly contributions. You can create and link your ORCID from the home page of the MTS by clicking on 'Modify my Springer Nature account'. For more information please visit www.springernature.com/orcid.

We hope to receive your revised paper within three weeks. If you cannot send it within this time, please let us know.

Best,
Fernando (on behalf of Jie Pan)

--

Fernando Chirigati, PhD
Chief Editor, Nature Computational Science
Nature Portfolio

Reviewers comments:

Reviewer #1 (Remarks to the Author):

In this manuscript, the authors have introduced MeDuSA, an algorithm that leverages single-cell RNA-seq data as a reference to estimate cell-state abundance in bulk RNA-seq data. They show that MeDuSA improves the estimation accuracy over the state-of-the-art methods by severalfold on average using simulation study. The author applied MeDuSA to in four case studies and demonstrate MeDuSA can give rise to deeper insights into disease etiology and biological mechanisms. Overall, this is a well-researched and well written manuscript. However, I have two minor comments:

1. The author compared the deconvolution accuracy of MeDuSA with the existing methods by the Pearson's correlation (R) and the root mean square deviation (RMSD) between the estimated cell-state abundance and the round truth (Fig. 1-3). Whether these existing methods are applied on the whole cell population, or the subset of the cell/cell types same as MeDuSA? it might be fair to compare on same cell subsets, since MeDuSA pre-selected cell states along trajectory for deconvolution.
2. Another related question. I noticed correlation (R) for MeDuSA can approach near 1 (Fig. 2b) , while the exiting methods in real data is even around 0 (Fig. 3a). While as the authors claimed in text "The result showed that MeDuSA outperformed CPM and the three cell-type deconvolution methods by severalfold", it's still a bit surprising on such big difference of performance? Is it because MeDuSA use trajectory information? Or any other explanations on it?

Reviewer #2 (Remarks to the Author):

In this study, Song et al have developed a linear mixed model able to estimate cell-state abundance in bulk RNA-seq data using single-cell RNA-seq data as reference. Major strengths of this work include: (i) source data and code (R-package) are publicly available, facilitating reproducibility and adoption of the method; (ii) novelty: the method leverages other cell states for the same cell type and use them as random effects to capture inter-cell-type variance across states.

Major comments:

About method comparison: I think it would be interesting to also compare the linear mixed model against a deep learning approach as hidden layers are represent higher-order latent representations that are robust to input noise and technical bias, see e.g., "Deep learning-based cell composition analysis from tissue expression profiles" by Menden et al. (Science 2020), "Deep autoencoder for interpretable tissue-adaptive deconvolution and cell-type-specific gene analysis" by Chen et al. (Nature Communications, 2022), etc.

Evaluation of statistical significance (p-values) of parameters of random effects component, as well fixed states, in the linear mixture model and note the fraction of significant genes. This would also corroborate the hypothesis that random effects (i.e., including other cell states) are significant and the better performance do not only rely on increased model complexity and may give insights about which

expressed genes from the same cell type benefit from this approach

Minor comments

About Figure 2: b-c) individual scores for different cell numbers are difficult to distinguish using color encoding, probably use different shapes instead of colored points, also better for color-blind people

Reviewer #2 (Remarks on code availability):

- 1) The code could benefit from more documentation.
- 2) A bash script is provided, which is good. However, something more proper would be to provide the pipeline as a snakemake or nextflow pipelines.

Reviewer #3 (Remarks to the Author):

In this manuscript, Song and colleagues present MeDuSA, a deconvolution method for inferring cell state abundance along a continuous linear trajectory from bulk expression data. Notably, this focus distinguishes MeDuSA from previous deconvolution tools that infer cell type/state abundances from bulk expression data without any explicit trajectory component. The authors perform several benchmarking analyses to evaluate their approach, mostly using simulation of pseudo-bulk samples from scRNA-seq data but also using real paired bulk and scRNA-seq data from 5 healthy human esophageal biopsies. They also demonstrate the use of MeDuSA in four case studies.

Strengths of this manuscript include a novel focus on trajectory inference from bulk expression data; explicit consideration of collinearity, which is a major confounding factor for deconvolution methods; and freely available code.

Weaknesses include an unconvincing and contrived benchmarking analysis, overstatements and embellishments, and a lack of key methodological details (below). More importantly, the need for this approach over simpler techniques has not been established. These issues, along with other serious technical and conceptual shortcomings, significantly dampen enthusiasm for this work.

Major Comments:

1. One of the most important problems with this work is the concept itself. There are many gene signatures that can easily capture the trajectory of one cell state to another – even individual marker genes can do this well, as the authors show in Figure 4c. Thus, it would be very straightforward to define scRNA-seq-based gene signatures of a developmental process that cover all distributions in Figure 2a, then apply those signatures to bulk data in order to study developmental processes using standard enrichment tools (e.g., single-sample GSEA). Unfortunately, no analyses in this work establish the benefit of MeDuSA over gene-based approaches.

2. The title and abstract of this paper are borderline-misleading. For example, the former states that MeDuSA is a method for “mixed model-based deconvolution of cell-state abundance,” making it appear that this approach extends beyond linear trajectories (it doesn’t). A much more informative title would be, “Medusa: mixed model-based deconvolution of cell-state abundance along a linear continuous trajectory”. Moreover, the abstract criticizes “established cell-state deconvolution methods” as being limited, misrepresenting the fact that most of these methods were developed for an entirely different goal – i.e., to infer discrete cell type/state abundances from bulk data *independent* of a trajectory. Indeed, one can easily argue that MeDuSA is also quite limited – i.e., limited in scope to a single linear trajectory, making it a complement rather than an alternative to existing tools. To remedy this, the authors need to carefully revise their manuscript to make it abundantly clear what MeDuSA does and does not do. All mentions of cell state deconvolution with MeDuSA need to be qualified to “cell state deconvolution along a single linear trajectory,” or similar. Overstatements such as “We show by extensive benchmark analyses that MeDuSA improves the estimation accuracy over the state-of-the-art methods by severalfold on average” and “Our study provides a high-accuracy and fine-resolution cell-state deconvolution method” are far too generous in scope and should be restricted to “single linear trajectories” in all instances.
3. Line 84: To compare against previous deconvolution methods, the authors repurposed them for cell state deconvolution by “dividing the cell-state trajectory into several bins.” While this might be justifiable, no further methodological details are provided by the authors, whether in the main text or methods. Details explaining how the bins were selected, how performance varied as a function of the number of bins, and how each of the deconvolution methods were parameterized and applied are entirely missing.
4. Performance metrics: It is unclear how the authors compared the output of MeDuSA to ground truth in their Pearson and RMSE assessments. No plots showing the actual output of MeDuSA on individual samples (ideally real bulk samples) are provided. Scatter plots showing the output for all deconvolved cell states along the linear trajectory vs. their ground truth expectations are needed to properly assess performance.
5. The benchmarking analysis of only 5 real tissue samples is wholly inadequate in both breadth and depth. Many additional real data samples across diverse tissue types with known cell-state abundances are needed to properly determine performance.
6. It appears that MeDuSA includes cell types as covariates in the model if they are not part of the trajectory – however it doesn’t actually output any estimates of abundance for them. Is this correct? Furthermore, does adjusting for these cell types even improve performance? What if they are ignored?
7. How should cell-state abundance inferred by MeDuSA be interpreted? Is it fractional abundance (i.e., ranging from 0 to 1 [this doesn’t seem to be the case from running the code]), relative abundance (e.g., unitless coefficients in a linear regression), or something else? Can the abundance of one cell state be directly compared with another as if they are true abundance measures?
8. Supplementary Fig. 10: It is already well known that pseudo-bulk scRNA-seq has a moderately strong correlation with bulk RNA-seq. A cross-correlation matrix comparing bulk RNA-seq on one axis with pseudo-bulk scRNA-seq on the other is needed to establish whether paired samples are the most highly correlated with each other.

Minor Comments:

1. Line 26: CTCs are a very common abbreviation for circulating tumor cells. For clarity, we suggest that the authors adjust or remove this abbreviation.
2. Line 35: Not "cell lines", cell subsets
3. Line 38: Not necessarily a "time-dependent manner"; "context-dependent" would be more accurate.
4. Line 48: No aspect of CPM is explicitly based on cell-state trajectories.
5. Line 66: "We show by extensive benchmark analyses based on real data" -> This is dishonest – simulations based on real data are not real anymore.
6. CIBERSORT is not the most recent version of this method – why did the authors ignore CIBERSORTx (Nat Biotechnol 2019), which is explicitly tailored to scRNA-seq-based deconvolution and performs batch correction? The method is publicly available.

Reviewer #3 (Remarks on code availability):

I ran the code via Code Ocean -- it installed and ran to completion on the provided example data without error. I did not assess whether the results of the paper are reproducible, as to my knowledge, there is no mechanism provided by the authors to do so. The documentation and examples are mostly clear, though I think instructions on how to best preprocess the expression data, including the trajectory, along with explicit examples of how to generate trajectory data, are needed.

Author Rebuttal to Initial comments

Responses to the Reviewers

We thank the reviewers for their constructive comments, which have greatly improved the manuscript. We have addressed all the reviewers' comments point-by-point below (in blue) in this document and made the corresponding changes in the manuscript files (highlighted in yellow). Here is a summary of the main changes:

1. We have added three additional sample-matched bulk RNA-seq and scRNA-seq datasets from human bone marrows (n=8), induced pluripotent stem cells (iPSCs, n=6), and human embryonic stem cells (hPSC, n=6), respectively, to the real-data benchmark analysis (**Figure 3a**). The conclusions remain unchanged.
2. We have included two deep learning-based deconvolution methods (Scaden and TAPE) and a gene enrichment-based method (ssGSEA) in the simulation study and real-data benchmark analysis. Our results consistently showed that MeDuSA outperformed all the compared methods (**Figures 2-3**).
3. We have added scatter plots that compare the estimated cell-state abundances with the ground truth expectations in the simulation study (**Supplementary Figure 2**) and with those estimated from scRNA-seq in the real-data benchmark analysis (**Supplementary Figure 14**).
4. We have added a paragraph (lines 547-578) that provides a detailed explanation of the parameters used for the deconvolution methods benchmarked in this study. Furthermore, we have thoroughly reviewed our text to avoid overstatements.
5. We have created a tutorial webpage to guide users in using MeDuSA and reproducing our benchmark analysis. Additionally, we have summarized the datasets used in this study (**Supplementary Tables 1-2**) and made the relevant code available on GitHub.

Reviewer #1

(Remarks to the Author)

In this manuscript, the authors have introduced MeDuSA, an algorithm that leverages single-cell RNA-seq data as a reference to estimate cell-state abundance in bulk RNA-seq data. They show that MeDuSA improves the estimation accuracy over the state-of-the-art methods by severalfold on average using simulation study. The author applied MeDuSA to in four case studies and demonstrate MeDuSA can give rise to deeper insights into disease etiology and biological mechanisms. Overall, this is a well-researched and well written manuscript. However, I have two minor comments:

Re: We thank the reviewer for the summary of our study and the positive remarks.

1. The author compared the deconvolution accuracy of MeDuSA with the existing methods by the Pearson's correlation (R) and the root mean square deviation (RMSD) between the estimated cell-state abundance and the round truth (Fig. 1-3). Whether these existing methods are applied on the whole cell population, or the subset of the cell/cell types same as MeDuSA? it might be fair to compare on same cell subsets, since MeDuSA pre-selected cell states along trajectory for deconvolution.

Re: In the simulation study, we selected a subset of cells along a predetermined trajectory (i.e., the focal cell type) from the scRNA-seq data to serve as the simulation source and deconvolution reference and used the RNA-seq data generated from the simulation source as the deconvolution target. To ensure a fair comparison, we compared MeDuSA to existing methods using exactly the same reference and target data. We have provided additional clarification on this in the revised manuscript (lines 550-551).

2. Another related question. I noticed correlation (R) for MeDuSA can approach near 1 (Fig. 2b), while the exiting methods in real data is even around 0 (Fig. 3a). While as the authors claimed in text "The result showed that MeDuSA outperformed CPM and the three cell-type deconvolution methods by severalfold", it's still a bit surprising on such big difference of performance? Is it because MeDuSA use trajectory information? Or any other explanations on it?

Re: All existing methods, except CPM, were developed for deconvoluting cell types. In order to improve the accuracy of cell-type deconvolution, most of these methods fit cell types jointly in the model. However, when such a joint model is applied to cell-state deconvolution, the cell states will affect each other due to the strong correlations among them, resulting in compromised performance (lines 118-122; **Supplementary Figures 6-7**). This was noted in a previous benchmark study (Sutton et al., Nature Communications, 2022), which identified high collinearity among brain cell subtypes as the main factor leading to reduced deconvolution accuracy.

MeDuSA addresses this collinearity problem by fitting the focal state as a fixed effect and the remaining cells of the same type as random effects in a linear mixed model. In addition, as noted in the manuscript (lines 112-118), fitting the remaining cells as random effects allows each cell to have a specific weight on bulk gene expression, resulting in better capturing of the variance in bulk gene expression and thereby a more precise estimate of the focal state in the fixed-effect term (i.e.,

improved deconvolution accuracy). Other reasons for MeDuSA's improved accuracy include fitting the mean expression levels of other cell types as fixed-effect covariates and smoothing the estimated abundances around neighboring cell states (lines 104-106, and 122-125).

In the revised manuscript, the deconvolution accuracy is measured by the concordance correlation coefficient (CCC), Pearson's correlation (R) and the root mean square deviation (RMSD) between the estimated cell-state abundance and the simulated ground truth in the simulation study (or the cell-state abundance estimated from the sample-matched scRNA-seq data in the real-data benchmark analysis). CCC is used as the primary measure of deconvolution accuracy because it is less sensitive to overweighted outliers than R and more interpretable than RMSD (lines 92-95). The conclusions remain unchanged.

Reviewer #2

(Remarks to the Author):

In this study, Song et al have developed a linear mixed model able to estimate cell-state abundance in bulk RNA-seq data using single-cell RNA-seq data as reference. Major strengths of this work include: (i) source data and code (R-package) are publicly available, facilitating reproducibility and adoption of the method; (ii) novelty: the method leverages other cell states for the same cell type and use them as random effects to capture inter-cell-type variance across states.

Re: We thank the reviewer for the summary of our study and the positive comments.

Major comments:

About method comparison: I think it would be interesting to also compare the linear mixed model against a deep learning approach as hidden layers are represent higher-order latent representations that are robust to input noise and technical bias, see e.g., "Deep learning-based cell composition analysis from tissue expression profiles" by Menden et al. (Science 2020), "Deep autoencoder for interpretable tissue-adaptive deconvolution and cell-type-specific gene analysis" by Chen et al. (Nature Communications, 2022), etc.

Re: We thank the reviewer for this suggestion. The two deep learning-based methods, Scaden (Menden et al., Science Advances, 2020) and TAPE (Chen et al., Nature Communications, 2022), were originally developed for cell-type deconvolution. We have repurposed them for cell-state

deconvolution using the binning strategy, as used in other cell-type deconvolution methods, and included them in the method comparison in both the simulation study and real-data benchmark analysis. We used the default parameters optimized by the authors. Specifically, for Scaden, the simulation step processed 500 cells per pseudo-bulk data and 5000 pseudo-samples in total, and the training step used a batch size of 128, a learning rate of 1.00E-4, and 5000 steps. For TAPE, we used a variance threshold of 0.98, a batch size of 128, and 5000 epochs. The results suggested that Scaden and TAPE performed similarly to the other cell-type deconvolution methods compared, and MeDuSA still outperformed all the compared methods by a substantial margin (**Figures 2-3**). We have updated the manuscript accordingly (lines 99-102, and 153-155).

Evaluation of statistical significance (p-values) of parameters of random effects component, as well fixed states, in the linear mixture model and note the fraction of significant genes. This would also corroborate the hypothesis that random effects (i.e., including other cell states) are significant and the better performance do not only rely on increased model complexity and may give insights about which expressed genes from the same cell type benefit from this approach.

Re: We thank the reviewer for this suggestion. We believe that this evaluation is better performed in the simulation study where the ground truth is known. We have reported the significance level, i.e., $-\log_{10}(\text{p-value})$, of the random-effect component from all simulations, which showed that the random-effect component was significant in all simulations (maximum $P = 1.35 \times 10^{-10}$), and the significance level decreased with fewer cells fitted (**Supplementary Figure 3**). We have updated the manuscript accordingly (lines 110-111). We have also included an option in our software tool to report p-values for the random-effect component.

We did not report the p-value for the fixed effect (i.e., the focal state) because a high significance level does not necessarily indicate high deconvolution accuracy. In fact, removing the random effects from the model would result in some of the effects will be absorbed by the focal cell state, leading to a more significant but upwardly biased estimate. As demonstrated in the manuscript (**Supplementary Figure 4**), the deconvolution accuracy of a model that only fits the focal state is substantially lower than that of MeDuSA.

The signature genes are selected before fitting the linear mixed model. Therefore, the significance level of the focal state or the random-effect component does not affect the selection of the signature genes.

Minor comments

About Figure 2: b-c) individual scores for different cell numbers are difficult to distinguish using color encoding, probably use different shapes instead of colored points, also better for color-blind people

Re: We have attempted to use different shapes to represent different cell numbers, but the resulting figure was not clear due to overlapping shapes (see **Figure R1**). Therefore, we have removed the cell number annotation from the main figure (**Figure 2b**) but still retain it in the supplementary figures (**Supplementary Figure 1**).

Figure R1. Benchmarking the deconvolution methods by simulation.

Reviewer #2 (Remarks on code availability):

- 1) The code could benefit from more documentation.

Re: We have updated the documentation of the MeDuSA code to provide more clarity and detail (<https://leonsong1995.github.io/MeDuSA/>).

2) A bash script is provided, which is good. However, something more proper would be to provide the pipeline as a snakemake or nextflow pipelines.

Re: Although we appreciate the suggestion to provide a snakemake or nextflow pipeline, we have instead developed a tutorial website to guide users in using MeDuSA and reproducing our benchmark analysis (<https://leonsong1995.github.io/MeDuSA/>).

Reviewer #3

(Remarks to the Author):

In this manuscript, Song and colleagues present MeDuSA, a deconvolution method for inferring cell-state abundance along a continuous linear trajectory from bulk expression data. Notably, this focus distinguishes MeDuSA from previous deconvolution tools that infer cell type/state abundances from bulk expression data without any explicit trajectory component. The authors perform several benchmarking analyses to evaluate their approach, mostly using simulation of pseudo-bulk samples from scRNA-seq data but also using real paired bulk and scRNA-seq data from 5 healthy human esophageal biopsies. They also demonstrate the use of MeDuSA in four case studies.

Strengths of this manuscript include a novel focus on trajectory inference from bulk expression data; explicit consideration of collinearity, which is a major confounding factor for deconvolution methods; and freely available code.

Weaknesses include an unconvincing and contrived benchmarking analysis, overstatements and embellishments, and a lack of key methodological details (below). More importantly, the need for this approach over simpler techniques has not been established. These issues, along with other serious technical and conceptual shortcomings, significantly dampen enthusiasm for this work.

Re: We thank the reviewer for the summary of our study and the insightful comments. We have addressed all the concerns point-by-point below.

Major Comments:

1. One of the most important problems with this work is the concept itself. There are many gene signatures that can easily capture the trajectory of one cell-state to another – even individual marker genes can do this well, as the authors show in Figure 4c. Thus, it would be very straightforward to define scRNA-seq-based gene signatures of a developmental process that cover all distributions in Figure 2a, then apply those signatures to bulk data in order to study developmental processes using standard enrichment tools (e.g., single-sample GSEA). Unfortunately, no analyses in this work establish the benefit of MeDuSA over gene-based approaches.

Re: We agree with the reviewer that gene enrichment tools such as ssGSEA can be used to estimate cell-type abundance from bulk RNA-seq data and can also be repurposed for cell-state deconvolution. However, a critical limitation of enrichment methods is that they do not consider the abundance of other cell types when estimating the abundance of the focal cell type. In bulk RNA-seq data, the expression level of signature genes of the focal cell type can be confounded by the expression levels of other cell types, leading to a biased estimate of cell abundance. To address this confounding effect, most cell-type deconvolution methods fit multiple cell types jointly. However, as noted in our response to a comment from reviewer #1, when the joint model is applied to cell-state deconvolution, the cell states can affect each other due to strong correlations among them, resulting in compromised performance (lines 118-122 and **Supplementary Figure 6**). MeDuSA addresses this issue by fitting the focal state as a fixed effect and cells in other states as random effects in a linear mixed model, which controls for the confounding effects (by the random-effect component) and ameliorates the collinearity problem (by splitting the focal state and other states as fixed and random effects separately) (**Supplementary Figure 7**).

We have compared MeDuSA to ssGSEA in the real-data benchmark analysis (see our response to comment #5 from this reviewer for a more detailed description of the improved real-data benchmark analysis). We used the "FindMarkers" function of the Seurat package to identify signature genes of each cell state (top 5 genes with the lowest p-values) and the GSEA package to compute enrichment scores. The results showed that MeDuSA still outperformed ssGSEA substantially for one-dimensional cell trajectories (**Figures 2-3**).

Although in some cases, ssGSEA showed a relatively high Pearson's correlation, the estimated cell-state abundance distribution along the predefined trajectory from ssGSEA did not agree well with that obtained from scRNA-seq data, as illustrated in **Figure R2a**. In light of this issue, we introduced

the Concordance Correlation Coefficient (CCC) (Lin et al., Biometrics, 1989), which evaluates the agreement between two variables with different units of measurement. We have opted to use CCC as the primary measure of deconvolution accuracy, as it is less sensitive to overweighted outliers than Pearson's correlation and more interpretable than RMSD (lines 91-95). In the simulation study, the mean CCC across all the scenarios and replicates was 0.86 for MeDuSA and 0.24 for ssGSEA (**Figure 2b**). In the real-data benchmark analysis, the mean CCC across all scenarios was 0.70 for MeDuSA, compared to 0.17 for ssGSEA (**Figures 2-3**).

Furthermore, we have explored the performance of ssGSEA in the real-data benchmark analysis using different numbers of signature genes. As we increased the number of signature genes, we observed a decreasing CCC and flattened estimates of cell-state abundance for ssGSEA (**Figures R2b-c**). This is likely due to decreased specificity of the signature genes, resulting in a higher level of contamination from other states. We have revised the manuscript accordingly (lines 91-95, 99-102, and 142-160).

Figure R2 Benchmarking MeDuSA and ssGSEA with real bulk RNA-seq data. Panel a shows the estimated cell-state abundances along a one-dimensional trajectory for both methods in two cases (see **Figure 3** for more details about the monocyte dataset). The abundances deconvoluted by ssGSEA are not consistent with those obtained from scRNA-seq data, despite the relatively high Pearson correlation coefficient (R). Panel b shows the cell-state abundances estimated by ssGSEA using 10 and 30 marker genes for each state. Panel c shows the concordance correlation coefficient (CCC) of ssGSEA in four real bulk RNA-seq datasets using different numbers of signature genes.

2. The title and abstract of this paper are borderline-misleading. For example, the former states that MeDuSA is a method for “mixed model-based deconvolution of cell-state abundance,” making it appear that this approach extends beyond linear trajectories (it doesn’t). A much more informative title would be, “Medusa: mixed model-based deconvolution of cell-state abundance along a linear continuous trajectory”. Moreover, the abstract criticizes “established cell-state deconvolution methods” as being limited, misrepresenting the fact that most of these methods were developed for an entirely different goal – i.e., to infer discrete cell type/state abundances from bulk data *independent* of a trajectory. Indeed, one can easily argue that MeDuSA is also quite limited – i.e., limited in scope to a single linear trajectory, making it a complement rather than an alternative to existing tools.

To remedy this, the authors need to carefully revise their manuscript to make it abundantly clear what MeDuSA does and does not do. All mentions of cell-state deconvolution with MeDuSA need to be qualified to “cell-state deconvolution along a single linear trajectory,” or similar. Overstatements such as “We show by extensive benchmark analyses that MeDuSA improves the estimation accuracy over the state-of-the-art methods by severalfold on average” and “Our study provides a high-accuracy and fine-resolution cell-state deconvolution method” are far too generous in scope and should be restricted to “single linear trajectories” in all instances.

Re: We thank the reviewer for the helpful comments. We have revised the title to read, “MeDuSA: mixed model-based deconvolution of cell-state abundances along a one-dimensional trajectory”. We opted not to use the word “linear” because a linear relationship is often interpreted as a straight line.

Regarding the comment on the statement about existing methods, we have revised the relevant text to read, “Deconvoluting cell-state abundances from bulk RNA-seq data can add considerable value to

existing data, but achieving fine-resolution and high-accuracy deconvolution remains a challenge (lines 11-12)."

Regarding the comment on overstatement, we have revised the text throughout the entire manuscript, wherever necessary, to emphasize that MeDuSA is a method for cell-state deconvolution across a one-dimensional trajectory. For example, we have defined specifically in the Abstract that MeDuSA is "a mixed model-based method that leverages single-cell RNA-seq data as a reference to estimate cell-state abundances along a one-dimensional trajectory in bulk RNA-seq data" and have revised the statements mentioned by the reviewer to read, "Extensive simulations and real-data benchmark analyses demonstrate that MeDuSA greatly improves the estimation accuracy over existing methods for one-dimensional trajectories" (lines 16-18) and "Our study provides a high-accuracy and fine-resolution method for cell-state deconvolution along a one-dimensional trajectory" (lines 20-23).

We have further made several changes to either improve or tone down the statements that relate to the reviewer's comments. Here are a few examples.

Lines 55-57: "While CPM has considerably improved the deconvolution resolution, the accuracy of the estimated cell-state abundance can still be improved".

Lines 60-63: "In this study, we introduce MeDuSA (mixed model-based deconvolution of cell-state abundance), a high-accuracy and fine-resolution cellular deconvolution method that leverages scRNA-seq data as a reference to estimate cell-state abundance along a one-dimensional trajectory in bulk RNA-seq data".

Lines 369-372: "Third, the cell-state trajectory modeled in the current version of MeDuSA is a one-dimensional vector, which may not fully portray the complexity of cellular transitions, particularly in cases of multiple cell trajectories. More work is warranted in the future to extend MeDuSA to model cell-states on a multi-dimensional space".

3. Line 84: To compare against previous deconvolution methods, the authors repurposed them for cell-state deconvolution by "dividing the cell-state trajectory into several bins." While this might be justifiable, no further methodological details are provided by the authors, whether in the main text or methods. Details explaining how the bins were selected, how performance varied as a function of the number of bins, and how each of the deconvolution methods were parameterized and applied are entirely missing.

Re: Regarding the binning strategy, we have included the following statements in the revised manuscript for clarification (lines 548-551): "We clustered cells into uniform cell-state bins along the predefined trajectory. To ensure a fair comparison between methods, we used the same cell-state bins in both MeDuSA and the compared methods. We provided the bin labels, reference data, and bulk RNA-seq data to the deconvolution methods to estimate cell-state abundances".

Regarding the performance of the deconvolution methods as a function of the number of bins, we previously examined MeDuSA and CPM with a range of cell-state bins through simulation (lines 125-128 and **Supplementary Figures 9-10**). Note that the cell-type deconvolution methods were not included in this analysis as they were developed to work with a limited number of cell types, and most of them failed to work when the number of bins was large. In response to the reviewer's comment, we have conducted additional benchmark analysis using real bulk RNA-seq data with varying numbers of cell states. To ensure that all methods were able to output abundance estimates, we set the number of states to 50, 100, 150, and 200, respectively. Consistent with the simulation results, MeDuSA was generally robust to the number of bins and remained the best-performing method in all scenarios (**Supplementary Figure 15**, lines 155-156).

Regarding the parameters of the deconvolution tools, we have included them in the revised manuscript as follows (lines 553-576):

- 1) For CPM, we downloaded the R package of scBio (v0.1.5) and used the default parameters of the CPM function, including "neighborhoodSize = 10, modelSize = 50, minSelection = 5". The resulting output of CPM provides abundance estimations for individual cells. To obtain estimates of cell-state abundance, we calculated the average abundance of cells within each cell-state bin.
- 2) For CIBERSORT, we downloaded the released R source code (v1.04) and built gene expression profiles (GEPs) by averaging gene expression profiles of cells in each cell-state bin. We used the signature genes selected by MeDuSA.
- 3) For BayesPrism, we downloaded the R package of BayesPrism (v2.0) and filtered out genes of chrX, chrY, mitochondria, and ribosome. We constructed the BayesPrism object using the function of "new.prism" with parameters of "outlier.cut=0.01, outlier.fraction=0.1" and used the functions of "run.prism" and "get.fraction" to obtain abundance estimations.
- 4) For Scaden, we downloaded the python package of scaden (v1.1.2) and generated pseudo-bulk data with settings of 500 cells per pseudo-bulk sample and 5,000 samples in total. Based on the

pseudo-bulk data, we trained the model using a batch size of 128, a learning rate of 1.00E-4, and 5,000 steps. We used the command of "predict" to obtain abundance estimations.

5) For TAPE, we downloaded the python package of TAPE (v1.1.0) and used the function of "Deconvolution" with parameters of "variance threshold=0.98, batch size=128, epochs=500".

6) For MuSiC, we downloaded the R package of MuSiC (v1.0.0) and ran it using the default settings. MuSiC states that it takes advantage of multiple subjects in the reference data to improve accuracy. In simulations, we randomly assigned datasets not generated from multiple subjects to two different subjects, as done in a previous study (Chen et al., Nature Communications, 2022). In the real-data benchmark analysis, we used reference scRNA-seq data generated from multiple donors and provided the donor labels to MuSiC.

7) For ssGSEA, we downloaded the R package of GSVA (v1.46) and used the function of "FindMarkers" of the Seurat package to identify signature genes (the top 5 genes with the lowest p-values) of each cell state.

4. Performance metrics: It is unclear how the authors compared the output of MeDuSA to ground truth in their Pearson and RMSE assessments. No plots showing the actual output of MeDuSA on individual samples (ideally real bulk samples) are provided. Scatter plots showing the output for all deconvolved cell states along the linear trajectory vs. their ground truth expectations are needed to properly assess performance.

Re: We appreciate the reviewer's suggestion and have included scatter plots for both the simulation study (**Supplementary Figure 2**) and real-data benchmark analysis (**Supplementary Figure 14**) in the revised manuscript. In the simulation study, we had four scenarios, 17 scRNA-seq datasets for each scenario, 5 replicates for each dataset, and 8 deconvolution methods for each simulation replicate. Due to space constraints, we pooled together the results from different datasets and simulation replicates into one scatter plot. For the real-data benchmark analysis, we had 4 datasets with varying numbers of samples and 8 deconvolution methods. Again, for space considerations, we pooled the results from different samples of each dataset into one scatter plot. In addition, as mentioned above, we have opted to use the concordance correlation coefficient (CCC) as the primary measure of deconvolution accuracy as it is less sensitive to overweighted outliers than Pearson's correlation (lines 91-95).

5. The benchmarking analysis of only 5 real tissue samples is wholly inadequate in both breadth and

depth. Many additional real data samples across diverse tissue types with known cell-state abundances are needed to properly determine performance.

Re: We thank the reviewer for this comment. First, we need to acknowledge that sample-matched bulk RNA-seq and scRNA-seq data are rarely available in the public domain. However, as per the reviewer's suggestion, we have managed to add 3 additional sample-matched bulk RNA-seq and scRNA-seq datasets (total $n = 37$) obtained from tissues or cell lines exhibiting well-defined cell states to the benchmark analysis (**Figures 3a-b, Supplementary Figures 13**). The details of the additional datasets are summarized below:

1) Human esophagus ($n=15$, Madisson et al., *Genome Biology*, 2020). In our previous manuscript, we used scRNA-seq and bulk RNA-seq data obtained from healthy, freshly harvested esophagi (the time point of 0 h, $n=5$). In the revised manuscript, we expanded our benchmark analysis to include data from time points 12 h and 24 h ($n=15$), as the quality of the scRNA-seq data remained stable during 24 hours of cold storage (Madisson et al., *Genome Biology*, 2020).

2) Human bone marrow ($n=8$, Oetjen et al., *JCI insight*, 2018). This dataset consists of human bone marrow samples collected from 8 donors. We used the monocytes development trajectory (from granulocyte-monocyte progenitors to non-classical monocytes) for benchmark analysis.

3) iPSC ($n=6$, Lappalainen et al., *Nature*, 2013; Cuomo et al., *Nature Communications*, 2020). Lappalainen et al. conducted bulk RNA-seq analysis on iPSCs, and Cuomo et al. generated scRNA-seq data using the same cell lines. Cuomo et al. (*Genome Biology*, 2021) showed a high level of consistency between the two datasets on day 0 of cultivation. Therefore, for the benchmark analysis, we utilized the scRNA-seq and bulk RNA-seq data from day 0 of cultivation. To ensure accurate measurements of cell-state abundance in the scRNA-seq data, we included samples with a minimum of 200 cells, resulting in the inclusion of six samples.

4) hPSC ($n=6$, Chu et al., *Genome Biology*, 2016). Chu et al. performed scRNA-seq and bulk RNA-seq of hPSCs, with the bulk RNA-seq data collected from multiple technical replicates. We used the mean of technical replicates to reduce noise in the data.

The benchmark analysis results from the additional datasets from different tissues, cell types, or time points consistently showed that MeDuSA outperformed existing methods in estimating cell-state abundances along a one-dimensional trajectory (**Figure 3c**). We have revised the manuscript accordingly (lines 142-160, and lines 604-625).

6. It appears that MeDuSA includes cell types as covariates in the model if they are not part of the trajectory – however it doesn't actually output any estimates of abundance for them. Is this correct? Furthermore, does adjusting for these cell types even improve performance? What if they are ignored?

Re: The reviewer is correct that MeDuSA does not output the estimated cell-type compositions, as the signature genes are selected to differentiate among different states of the focal cell type rather than cell types. However, this does not preclude associations between the bulk expression level of the signature genes and cell-type compositions. Such a confounding effect, if not modeled, can have a substantial impact on the accuracy of cell-state deconvolution, especially when the composition of the other cell types is large. This is demonstrated by our additional simulation analysis in which we varied the composition of the other cell types from 0 to 0.9. The performance of MeDuSA remained stable with the change of the composition of the other cell types, whereas removing the cell type covariates from the MeDuSA model led to decreased deconvolution accuracy, and such a decrease increased with the increase of the composition of the other cell types (**Supplementary Figure 10**). We have updated the manuscript accordingly (lines 128-130).

7. How should cell-state abundance inferred by MeDuSA be interpreted? Is it fractional abundance (i.e., ranging from 0 to 1 [this doesn't seem to be the case from running the code]), relative abundance (e.g., unitless coefficients in a linear regression), or something else? Can the abundance of one cell-state be directly compared with another as if they are true abundance measures?

Re: The parameters that MeDuSA seeks to estimate are the fractional abundances of different cell states within the focal cell type, which are bound between 0 and 1 and sum up to unity. It is noteworthy that since we estimate these parameters from a linear model, to ensure unbiased estimation, the estimates are not constrained so that their estimates can be out of the parameter space, and the sum of the estimates can deviate from unity. Such an abundance estimate can be compared across cell states, as demonstrated in our applications of esophagus tumors and COVID-19, or across individuals, as demonstrated in our applications of melanoma and *csd*-eQTLs. For ease of interpretation, one can rescale the raw abundance estimates to range from 0 to 1 and sum up to unity.

We have clarified this in the revised manuscript (lines 444-448).

8. Supplementary Fig. 10: It is already well known that pseudo-bulk scRNA-seq has a moderately strong correlation with bulk RNA-seq. A cross-correlation matrix comparing bulk RNA-seq on one axis with pseudo-bulk scRNA-seq on the other is needed to establish whether paired samples are the most highly correlated with each other.

Re: In **Supplementary Figure 16**, we have included a heatmap of the cross-correlation matrix between the actual bulk RNA-seq data and the pseudo-bulk scRNA-seq data.

Minor Comments:

1. Line 26: CTCs are a very common abbreviation for circulating tumor cells. For clarity, we suggest that the authors adjust or remove this abbreviation.

Re: To avoid confusion, we have eliminated this abbreviation from the revised manuscript (see lines 28-29).

2. Line 35: Not “cell lines”, cell subsets

Re: We have revised the text accordingly (line 39).

3. Line 38: Not necessarily a “time-dependent manner”; “context-dependent” would be more accurate.

Re: We have revised the text accordingly (line 42).

4. Line 48: No aspect of CPM is explicitly based on cell-state trajectories.

Re: The CPM authors used the term “cell-state space” (see the text below from the “Overview of CPM” section of the CPM paper).

“We developed CPM, a method based on computational deconvolution for identifying a cell population map from bulk gene expression data of a heterogeneous sample. In our framework, the cell population map is the abundance of cells over a cell-state space.”

To maintain consistency with the definition/terminology used in the CPM paper, we have revised the relevant text to read (lines 51-55), "Cell Population Mapping (CPM) is a cellular deconvolution method specifically designed to exploit a "cell-state space" inferred from reference scRNA-seq data to estimate cell-state abundances in bulk RNA-seq data. CPM partitions the cell-state space into several grids, constructs a GEP by randomly sampling a cell from each grid, and combines the estimated abundances across thousands of repeats to obtain a single abundance for each cell".

5. Line 66: "We show by extensive benchmark analyses based on real data" -> This is dishonest – simulations based on real data are not real anymore.

Re: We have revised the text to read, "We show by extensive simulations and real-data benchmark analyses that MeDuSA is substantially more accurate than existing methods when assessed with one-dimensional trajectories" (lines 70-72).

6. CIBERSORT is not the most recent version of this method – why did the authors ignore CIBERSORTx (Nat Biotechnol 2019), which is explicitly tailored to scRNA-seq-based deconvolution and performs batch correction? The method is publicly available.

Re: We appreciate the suggestion made by the reviewer. However, we found that the CIBERSORTx web tool has a limit in terms of the size of the reference scRNA-seq data, which renders it unsuitable for most of our simulation and real-data benchmark analyses.

In response, we have included four additional cell type deconvolution methods (MuSiC, BayesPrism, Scaden, and TAPE) that were specifically designed for scRNA-seq-based deconvolution. Each of these methods has a unique approach for correcting technical batch effects between scRNA-seq and bulk RNA-seq datasets.

Reviewer #3 (Remarks on code availability):

I ran the code via Code Ocean -- it installed and ran to completion on the provided example data without error. I did not assess whether the results of the paper are reproducible, as to my knowledge, there is no mechanism provided by the authors to do so. The documentation and examples are mostly clear, though I think instructions on how to best preprocess the expression data, including the trajectory, along with explicit examples of how to generate trajectory data, are needed.

Re: We have provided a tutorial web to guide users in utilizing MeDuSA and replicating our benchmark analysis. Additionally, we have provided a summary of the data access for the experiments we conducted (**Supplementary Tables 1-2**) and made the relevant code available on GitHub (<https://leonsong1995.github.io/MeDuSA/>).

Decision Letter, first revision:

Date: 15th May 23 23:56:13
Last Sent: 15th May 23 23:56:13
Triggered By: Jie Pan
From: jie.pan@us.nature.com
To: jian.yang@westlake.edu.cn
CC: computacionalscience@nature.com
BCC: jie.pan@us.nature.com
Subject: AIP Decision on Manuscript NATCOMPUTSCI-22-1257A
Message: Our ref: NATCOMPUTSCI-22-1257A

15th May 2023

Dear Dr. Yang,

Thank you for submitting your revised manuscript "MeDuSA: mixed model-based deconvolution of cell-state abundances along a one-dimensional trajectory" (NATCOMPUTSCI-22-1257A). It has now been seen by the original referees and their comments are below (sorry for the delay - since one of the original referees dropped out in this round). The reviewers find that the paper has improved in revision, and therefore we'll be happy in principle to publish it in Nature Computational Science, pending minor revisions to satisfy the referees' final requests and to comply with our editorial and formatting guidelines.

TRANSPARENT PEER REVIEW

Nature Computational Science offers a transparent peer review option for original research manuscripts. We encourage increased transparency in peer review by publishing the reviewer comments, author rebuttal letters and editorial decision letters if the authors agree. Such peer review material is made available as a supplementary peer review file. **Please remember to choose, using the manuscript system, whether or not you want to participate in transparent peer review.**

Please note: we allow redactions to authors' rebuttal and reviewer comments in the interest of confidentiality. If you are concerned about the release of confidential data, please let us know specifically what information you would like to have removed. Please note that we cannot incorporate redactions for any other reasons. Reviewer names will be published in the peer review files if the reviewer signed the comments to authors, or if reviewers explicitly agree to release their name. For more information, please refer to our <https://www.nature.com/documents/nr->

transparent-peer-review.pdf" target="new">FAQ page.

Thank you again for your interest in Nature Computational Science. Please do not hesitate to contact me if you have any questions.

Sincerely,

Jie Pan, Ph.D.
Senior Editor
Nature Computational Science

ORCID

Reviewer #1 (Remarks to the Author):

The points I raised in the previous round of review have been satisfactorily addressed. I believe the paper is now acceptable for publication.

Reviewer #3 (Remarks to the Author):

The revised manuscript by Song and colleagues is much improved and my comments have been adequately addressed. My only remaining concern is the use of Lin's CCC for evaluating the output of ssGSEA, as the latter does not produce fractional abundances, but rather enrichment scores. Since CCC is intended to compare measurements of the same variable, it is unreasonable to apply CCC to compare ssGSEA results against ground truth fractions.

Another minor concern is related to comment #1 from the "Remarks on code availability" from R2, in which additional documentation was requested. I would like the authors to do a better job emphasizing that MeDuSA is restricted to inferring one-dimensional trajectories from bulk data in their descriptions of MeDuSA available online (<https://leonsong1995.github.io/MeDuSA/> and <https://github.com/LeonSong1995/MeDuSA/>). Currently, they emphasize "cell-state abundance" estimation, which is too generous.

Reviewer #3 (Remarks on code availability):

The authors have added appropriate documentation and examples in response to my comment.

Author Rebuttal, first revision:

Responses to the Reviewers

Reviewer #1

(Remarks to the Author):

The points I raised in the previous round of review have been satisfactorily addressed. I believe the paper is now acceptable for publication.

Re: We thank the reviewer for their valuable input throughout the review process.

Reviewer #3

(Remarks to the Author):

The revised manuscript by Song and colleagues is much improved and my comments have been adequately addressed.

Re: We appreciate the reviewer's positive evaluation of our revised manuscript and their recognition of the improvements made in response to their insightful comments.

My only remaining concern is the use of Lin's CCC for evaluating the output of ssGSEA, as the latter does not produce fractional abundances, but rather enrichment scores. Since CCC is intended to compare measurements of the same variable, it is unreasonable to apply CCC to compare ssGSEA results against ground truth fractions.

Re: The reviewer is correct in noting that it is inappropriate to apply Lin's CCC to raw ssGSEA scores. In our previous manuscript, we had already scaled the ssGSEA scores to fractional abundances ranging from 0 to 1. Using raw ssGSEA scores would have resulted in even lower Lin's CCC values (see **Figure R1**). In the revised manuscript, we have clarified that to ensure a fair comparison in the benchmark analysis, the ssGSEA output scores were scaled to fractional abundances between 0 and 1 (Section 3 of the Supplementary Note: Point 7).

Figure R1. Boxplot of CCC values using raw ssGSEA scores in the benchmark analysis.

Another minor concern is related to comment #1 from the "Remarks on code availability" from R2, in which additional documentation was requested. I would like the authors to do a better job emphasizing that MeDuSA is restricted to inferring one-dimensional trajectories from bulk data in their descriptions of MeDuSA available online (<https://leonsong1995.github.io/MeDuSA/> and <https://github.com/LeonSong1995/MeDuSA/>). Currently, they emphasize "cell-state abundance" estimation, which is too generous.

Re: We have revised the text throughout the online tutorial, as needed, to emphasize that MeDuSA is a method for cell-state deconvolution across a one-dimensional trajectory.

Reviewer #3

(Remarks on code availability):

The authors have added appropriate documentation and examples in response to my comment.

Re: We thank the reviewer for their valuable input throughout the review process.

Final Decision Letter:**Date:** 13th June 23 13:12:08**Last Sent:** 13th June 23 13:12:08**Triggered By:** Jie Pan**From:** jie.pan@us.nature.com**To:** jian.yang@westlake.edu.cn**BCC:** fernando.chirigati@us.nature.com,rjsart@springernature.com,rjsproduction@springernature.com,jie.pan@us.nature.com,computationalscience@nature.com**Subject:** Decision on Nature Computational Science manuscript NATCOMPUTSCI-22-1257B**Message:** Dear Professor Yang,

We are pleased to inform you that your Article "Mixed model-based deconvolution of cell-state abundances (MeDuSA) along a one-dimensional trajectory" has now been accepted for publication in Nature Computational Science.

Once your manuscript is typeset, you will receive an email with a link to choose the appropriate publishing options for your paper and our Author Services team will be in touch regarding any additional information that may be required.

Please note that *Nature Computational Science* is a Transformative Journal (TJ). Authors may publish their research with us through the traditional subscription access route or make their paper immediately open access through payment of an article-processing charge (APC). Authors will not be required to make a final decision about access to their article until it has been accepted. [Find out more about Transformative Journals](https://www.springernature.com/gp/open-research/transformative-journals)

Acceptance of your manuscript is conditional on all authors' agreement with our publication policies (see <https://www.nature.com/natcomputsci/for-authors>). In particular your manuscript must not be published elsewhere and there must be no announcement of the work to any media outlet until the publication date (the day on which it is uploaded onto our web site).

Before your manuscript is typeset, we will edit the text to ensure it is intelligible to our wide readership and conforms to house style. We look particularly carefully at the titles of all papers to ensure that they are relatively brief and understandable.

Once your manuscript is typeset and you have completed the appropriate grant of rights, you will receive a link to your electronic proof via email with a request to make any corrections within 48 hours. If, when you receive your proof, you cannot meet this deadline, please inform us at rjsproduction@springernature.com immediately.

If you have queries at any point during the production process then please contact the production team at rjsproduction@springernature.com. Once your paper has been scheduled for online publication, the Nature press office will be in touch to confirm the details.

Content is published online weekly on Mondays and Thursdays, and the embargo is set at 16:00 London time (GMT)/11:00 am US Eastern time (EST) on the day of publication. If you need to know the exact publication date or when the news embargo will be lifted, please contact our press office after you have submitted your proof corrections. Now is the time to inform your Public Relations or Press Office about your paper, as they might be interested in promoting its publication. This will allow them time to prepare an accurate and satisfactory press release. Include your manuscript tracking number NATCOMPUTSCI-22-1257B and the name of the journal, which they will need when they contact our office.

About one week before your paper is published online, we shall be distributing a press release to news organizations worldwide, which may include details of your work. We are happy for your institution or funding agency to prepare its own press release, but it must mention the embargo date and Nature Computational Science. Our Press Office will contact you closer to the time of publication, but if you or your Press Office have any inquiries in the meantime, please contact press@nature.com.

We welcome the submission of potential cover material (including a short caption of around 40 words) related to your manuscript; suggestions should be sent to Nature Computational Science as electronic files (the image should be 300 dpi at 210 x 297 mm in either TIFF or JPEG format). We also welcome suggestions for the Hero Image, which appears at the top of our <http://www.nature.com/natcomputsci> home page; these should be

72 dpi at 1400 x 400 pixels in JPEG format. Please note that such pictures should be selected more for their aesthetic appeal than for their scientific content, and that colour images work better than black and white or grayscale images. Please do not try to design a cover with the Nature Computational Science logo etc., and please do not submit composites of images related to your work. I am sure you will understand that we cannot make any promise as to whether any of your suggestions might be selected for the cover of the journal.

Best regards,

Jie Pan, Ph.D.
Senior Editor
Nature Computational Science

P.S. Click on the following link if you would like to recommend Nature Computational Science to your librarian: <https://www.springernature.com/gp/librarians/recommend-to-your-library>

** Visit the Springer Nature Editorial and Publishing website at <http://editorial-jobs.springernature.com> for more information about our career opportunities. If you have any questions please click [here](mailto:editorial.publishing.jobs@springernature.com).**